# IMTS is Worth Time × Channel Patches: Visual Masked Autoencoders for Irregular Multivariate Time Series Prediction

**Zhangyi Hu** [* 1]   **Jiemin Wu** [* 1]   **Hua Xu** [* 1]   **Mingqian Liao** [1]   **Ninghui Feng** [1]   **Bo Gao** [2]   **Songning Lai** [1]
**Yutao Yue** [† 1 3]

## Abstract

Irregular Multivariate Time Series (IMTS) forecasting is challenging due to the unaligned nature of multi-channel signals and the prevalence of extensive missing data. Existing methods struggle to capture reliable temporal patterns from such data due to significant missing values. While pre-trained foundation models show potential for addressing these challenges, they are typically designed for Regularly Sampled Time Series (RTS). Motivated by the visual Mask AutoEncoder's (MAE) powerful capability for modeling sparse multi-channel information and its success in RTS forecasting, we propose **VIMTS**, a framework adapting **Vi**sual MAE for **IMTS** forecasting. To mitigate the effect of missing values, VIMTS first processes IMTS along the timeline into feature patches at equal intervals. These patches are then complemented using learned cross-channel dependencies. Then it leverages visual MAE's capability in handling sparse multichannel data for patch reconstruction, followed by a coarse-to-fine technique to generate precise predictions from focused contexts. In addition, we integrate self-supervised learning for improved IMTS modeling by adapting the visual MAE to IMTS data. Extensive experiments demonstrate VIMTS's superior performance and few-shot capability, advancing the application of visual foundation models in more general time series tasks. Our code is available at https://github.com/WHU-HZY/VIMTS.

## 1. Introduction

Irregular Multivariate Time Series (IMTS)(Weerakody et al., 2021) forecasting plays a crucial role in various domains, including finance (Bai & Ng, 2008), healthcare (Esteban et al.,

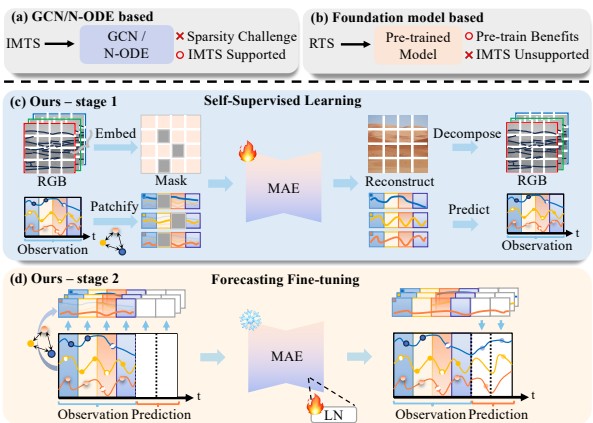

*Figure 1.* Illustration of our idea: **(a)** Current IMTS-specific methods struggle to capture reliable temporal patterns from such data due to significant missing values. **(b)** Pre-trained models show potential for modeling sparse data, but are limited to RTS. In contrast, as illustrated in (c) and (d), VIMTS segments data into time-aligned patches and imputes missing values at the representation level using time × channel patchify. It then leverages self-supervised learning to adapt the visual MAE's pre-trained capability for handling semantically sparse multi-channel data to IMTS data, leading to powerful performance and few-shot capability.

2017), transportation (Gong et al., 2021), and meteorology (Das & Ghosh, 2017). However, unlike structured data such as images or text, the semantic information in IMTS is embedded in complex dynamics across multiple channels over time, which are disrupted by irregular sampling and missing values. These challenges arise from various factors, including the randomness of monitored subjects, the reliability issues of data collection devices, and privacy concerns (Wang et al., 2024), thus complicating downstream tasks such as traffic flow forecasting and weather forecasting.

Early methods involve statistical imputation methods for synchronizing timestamps (Hamzaçebi, 2008; Van Buuren & Groothuis-Oudshoorn, 2011), but they require a deep understanding of system dynamics and inadvertently discard information contained in missing points (Horn et al., 2020). Although recent GCN-based methods (Zhang et al., 2024a) and Neural-ODE-based methods (Chen et al., 2018; De Brouwer et al., 2019; Rubanova et al., 2019; Schirmer et al., 2022) show progress in modeling cross-channel dependency and temporal dependencies of irregular samples,

---

*Equal contribution [1]The Hong Kong University of Science and Technology (Guangzhou), Guangzhou 511400, China [2]Beijing University of Posts and Telecommunications, Beijing, China [3]Institute of Deep Perception Technology, JITRI, Wuxi 214000, China. Correspondence to: Yutao Yue <yutaoyue@hkust-gz.edu.cn>.

*Proceedings of the 42nd International Conference on Machine Learning*, Vancouver, Canada. PMLR 267, 2025. Copyright 2025 by the author(s).

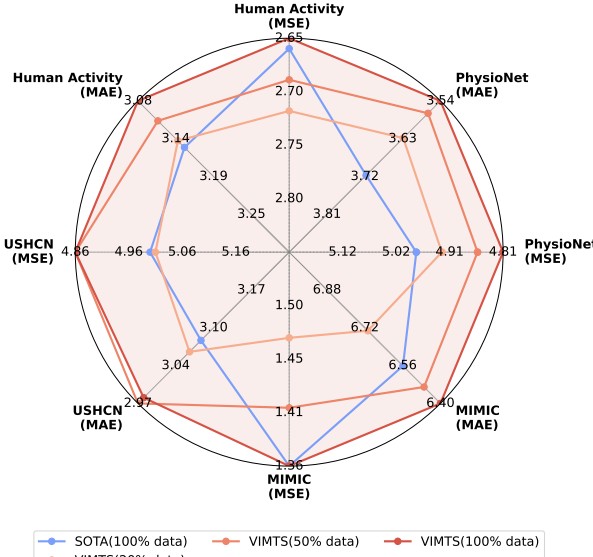

*Figure 2.* The illustration highlights VIMTS's superior Mean Absolute Error (MAE) and Mean Squared Error (MSE) relative to state-of-the-art methods on the PhysioNet, Human Activity, USHCN, and MIMIC datasets. Moreover, VIMTS maintains competitive performance in few-shot scenarios.

GCN-based methods alternately model temporal and channel information, causing severe cumulative error due to the sparsity, while N-ODE-based methods struggle to construct accurate models from some individual channels with significant missing values and simultaneously require substantial computational resources. These challenges lead to unreliable pattern capturing and poor few-shot capability.

In parallel, foundation models have revolutionized various areas (Jin et al., 2024; Das et al., 2024; Brown et al., 2020; He et al., 2022) by leveraging capability benefiting from large-scale pre-training to capture important features for downstream tasks with limited fine-tuning data. VisionTS (Chen et al., 2025) demonstrates that visual Mask AutoEncoders (MAEs) pre-trained on large-scale RGB images are naturally adaptable to time series data, as they share pattern similarities with natural images in terms of information density, and multichannel patterns. This suggests significant potential for applying visual foundation models to time series forecasting. Nevertheless, most existing pre-trained model based methods are designed for RTS data, limiting their application in more general and practical scenarios.

As illustrated in Fig. 1, motivated by the capabilities of visual MAEs in modeling semantically sparse multichannel information and their adaptability to the time series domain, we introduce a pioneering framework that leverages **V**isual pre-trained MAE for **IMTS** forecasting (VIMTS). The core idea is to adapt the powerful capabilities of pre-trained visual MAEs (He et al., 2022) to IMTS data via self-supervised latent space mask reconstruction for enhanced performance and few-shot capability. Specifically, VIMTS

treats IMTS as a time × channel image-like structure. It divides the data into sections along the timeline at equal intervals and employs a Transformable Time-aware Convolutional Network (TTCN) to extract intra-section feature patches. This addresses unstructured inputs and temporal misalignment. These patches that suffer from missing values are then complemented at the feature-level using cross-channel dependencies learned by Graph Convolutional Networks (GCNs) (Kipf & Welling, 2016). These complemented patches are then fed into a pre-trained visual MAE for understanding and reconstruction. This process models temporal dependencies for patches within each channel. Finally, a coarse-to-fine technique generates precise predictions by querying patch-level time period representations using their corresponding timestamps, thereby focusing on relevant temporal-channel context. To fully leverage the potential of the visual MAE and fully utilize historical data, we develop a two-stage training strategy. First, self-supervised learning is employed to improve IMTS modeling through adapting visual MAE to IMTS data. Second, supervised fine-tuning are utilized for more precise prediction. This strategy leads to significant improvement in forecasting accuracy and robust few-shot capability. Our main contributions include:

- We introduce VIMTS, a pioneering framework that leverages the powerful capability of visual MAE in modeling semantically sparse multichannel data for IMTS forecasting. As shown in Fig. 2, extensive experiments on four real-world datasets demonstrate its superior performance compared to existing baselines and its robust few-shot capability, paving the way for applying visual foundation models to more general time series forecasting tasks.

- We propose a new encoding-decoding strategy. For encoding, IMTS is processed into time-aligned feature patches along the timeline at equal intervals, which is then compensated with cross-channel information to mitigate the effect of missing values. For decoding, a coarse-to-fine strategy progressively generates predictions from patches to specific time points, focusing on related temporal-channel contexts for enhanced accuracy.

- We develop a two-stage training strategy. First, VIMTS employs self-supervised learning to improve IMTS modeling by adapting the capabilities of visual MAEs to IMTS data. Second, supervised fine-tuning is proposed for task-specific adaptation. This strategy leads to significant performance improvement and robust few-shot capability.

## 2. Methodology

### 2.1. Overview

The overall methodology is illustrated in Fig. 3. The architecture of VIMTS consists of three main components: time×channel patchify, time-wise reconstruction, and patch2point prediction. We employ a two-stage training strategy that encompasses self-supervised learning and supervised fine-tuning. In the following sections, we will

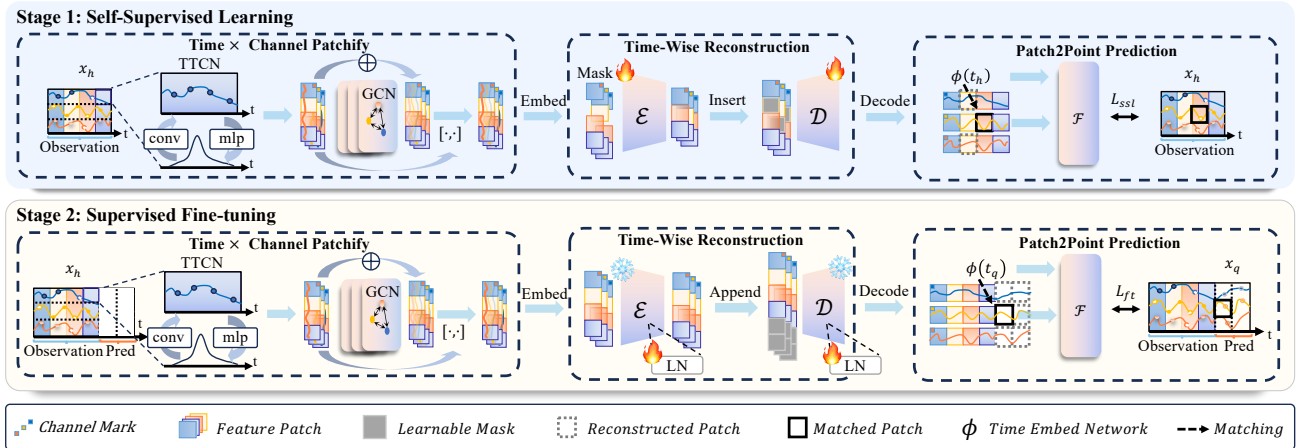

**Figure 3.** The overall architecture of VIMTS. The irregularly sampled data in each channel is divided into sections with equal-intervals along the timeline. Each section undergoes intra-section feature extraction using Time-aware Convolutional Network (TTCN) and cross-channel information compensation via Graph Convolutional Networks (GCNs). These compensated patches are then fed into a pre-trained MAE for patch reconstruction, thereby modeling temporal dependencies among patches within each channel. Finally, a coarse-to-fine technique gradually generates precise predictions from patch-level to point-level. The training encompasses two stages. First, self-supervised learning aims to improve IMTS modeling by adapting the capabilities of the visual pre-trained MAE to IMTS data. Second, the supervised fine-tuning is employed to enhance forecasting performance.

introduce each part of our pipeline in detail.

## 2.2. Task Definition

**IMTS Observation.** IMTS data with $N$ variables is represented by the triplet $\mathcal{O} = (\mathcal{T}, \mathcal{X}, \mathcal{M})$. Here, $\mathcal{T} = [t_l]_{l=1}^L \in \mathbb{R}^L$ contains $L$ unique timestamps, while matrix $\mathcal{X} = [[x_l^n]_{n=1}^N]_{l=1}^L \in \mathbb{R}^{L \times N}$ records observed values $x_l^n$ at the $l$-th timestamp $t_l$ for the $n$-th variable or 'NA' if unobserved. The mask matrix $\mathcal{M} = [[m_l^n]_{n=1}^N]_{l=1}^L \in \{0,1\}^{L \times N}$ indicates the availability of observation at the $l$-th timestamp $t_l$ for the $n$-th variable with $m_l^n = 1$, otherwise $m_l^n = 0$.

**IMTS Forecasting.** The task is to develop a model $\Theta$ that, given historical observations $\mathcal{O}$ and future query timestamps $\mathcal{Q} = \{[q_j^n]_{j=1}^{Q_n}\}_{n=1}^N$, where $q_j^n$ denotes that the $j$-th query timestamp of the $\mathcal{Q}_n$ queries for the $n$-th variable, to forecast the corresponding target values $\hat{\mathcal{X}} = \{[\hat{x}_j^n]_{j=1}^{Q_n}\}_{n=1}^N$, where $\hat{x}_j^n$ denotes the ground truth value at the query timestamp $q_j^n$. This process is represented as $\Theta(\mathcal{O}, \mathcal{Q}) \to \hat{\mathcal{X}}$.

## 2.3. Time × Channel Patchify

This section aims to transform unstructured IMTS data into patches while mitigating the effect of missing values. To achieve this, the module first processes IMTS into feature patches along the timeline at equal time intervals. Each time interval contains patches from all channels. These patches that suffer from missing values are then complemented by information from related channels according to the learned cross-channel dependencies. This process enables more reliable inter-patch temporal dependency modeling in MAE.

### 2.3.1. TIME-WISE DIVIDING AND EMBEDDING

In our method, an IMTS dataset $\mathcal{O}$ is divided into $P$ patches using uniform time windows of size $s$. Each patch $p$, for

$1 \le p \le P$, spans from $t_{start}^p$ to $t_{start}^p + s$, where $t_{start}^p = t_1 + (p-1)s$, and $t_1$ is the initial time. This approach ensures section-level temporal alignment and preserves the multichannel structure (Zhang et al., 2024a).

After dividing, we utilize learnable time embeddings to capture temporal patterns and encode continuous time information (Shukla & Marlin, 2021a). For a given timestamp $t$, its embedding $\phi(t)$ is defined as:

$$\phi(t)[d] = \begin{cases} \omega_0 \cdot t + \alpha_0, & \text{if } d = 0 \\ \sin(\omega_d \cdot t + \alpha_d), & \text{if } 0 < d < D_{te} \end{cases}, \quad (1)$$

where $\omega_d$ and $\alpha_d$ are learnable parameters and $D_{te}$ is the embedding dimension. These embeddings combine linear and periodic terms to capture non-periodic and periodic temporal patterns.

### 2.3.2. TEMPORAL FEATURE EXTRACTION

After time-wise dividing and embedding, we employ a Transformable Time-aware Convolutional Network (TTCN) to process variable-length sequences within time intervals (Zhang et al., 2024b) into patches with aligned shapes and semantics.

In detail, for the $n$-th channel, we concatenate the time embeddings $\phi(t_i^n)$ and the observation $x_i^n$ within the $p$-th time section:

$$\mathbf{x}_p^n = [\phi(t_i^n) \| x_i^n]_{i=l_p}^{r_p}, \quad t_i^n \in [t_{start}^p, t_{start}^p + s), \quad (2)$$

where $\|$ denotes the 'concatenate' operation, $l_p$ and $r_p$ denote the start and end index of the observation respectively within the $p$-th time section in the $n$-th channel.

TTCN then captures intra-section information within each channel by employing adaptive convolution filters:

$$\mathbf{f}_d^n = \left[ \frac{\exp(\mathbf{F}_d(\mathbf{x}_p^n[i]))}{\sum_{j=1}^{L_p} \exp(\mathbf{F}_d(\mathbf{x}_p^n[j]))} \right]_{i=1}^{L_p}, \qquad (3)$$

where, for the $n$-th channel, $L_p = l_p - r_p + 1$ is the number of points within the $p$-th time section, $\mathbf{f}_d^n \in \mathbb{R}^{L_p \times D_{in}}$ represents the filter for the $d$-th feature map, $D_{in}$ is the number of filters, and $\mathbf{F}_d$ denotes the $d$-th meta-filter (mlp).

With $D_{in}$ filters derived based on Eq. 3, we attain the $p$-th feature patch in the $n$-th channel $h_p^{n'} \in \mathbb{R}^{D_{in}}$ by the following temporal convolution:

$$h_p^{n'} = \left[ \sum_{i=1}^{L_p} \mathbf{f}_d^n[i]^\top \mathbf{x}_p^n[i] \right]_{d=1}^{D_{in}}. \qquad (4)$$

To handle sparse IMTS data, we enhance representations by concatenating a binary mask indicating the availability:

$$h_p^{n,m} = [h_p^{n'} \| m_p] \in \mathbb{R}^D, \qquad (5)$$

where $D = D_{in} + 1$, $m_p^n = 1$ indicates the presence of observations while $m_p^n = 0$ indicates an empty patch, $h_p^{n,m}$ denotes the feature patch concatenated with the mask indicator.

We further incorporate channel-specific embeddings to capture channel-specific traits (e.g., units, stats, missing patterns), thereby distinguishing heterogeneous channels to enhance the following cross-channel compensation and inter-patch temporal modeling within channels. In detail, for the $n$-th channel, we define learnable embeddings $e_n \in \mathbb{R}^D$, which is added to the feature patches to create the patches $h_p^n$ for cross-channel dependency modeling:

$$h_p^n = h_p^{n,m} + e_n. \qquad (6)$$

2.3.3. CROSS-CHANNEL INFORMATION INTERACTION

Due to extensive missing values in IMTS, patches of individual channels contain insufficient information for reliable temporal dependency modeling. To mitigate this, we employ GCN to model bidirectional channel dependencies and enrich each channel's representation with complementary information from correlated channels.

Inspired by (Zhang et al., 2024a), to learn bidirectional channel dependency graphs, we fuse static channel characteristics with dynamic patch features to create graph vertex embeddings. In the beginning, maintain two learnable embedding dictionaries $\mathbf{E}_1^s, \mathbf{E}_2^s \in \mathbb{R}^{N \times D_{ve}}$ which encode static characteristics (e.g., representing inflow/outflow nodes). They are then updated with dynamic patch information by a gated mechanism to obtain hybrid embeddings:

$$\mathbf{E}_{p,k} = \mathbf{E}_k^s + g_{p,k} \odot \mathbf{H}_p \mathbf{W}_k^d, \quad k \in \{1, 2\}, \qquad (7)$$

where $g_{p,k} = \mathrm{ReLU}(\tanh([\mathbf{H}_p \| \mathbf{E}_k^s] \mathbf{W}_k^g))$ controls the fusion of static and dynamic information, $\mathbf{H}_p = [h_p^n]_{n=1}^N \in$

$\mathbb{R}^{N \times D}$ denotes the $h_p^n$ concatenation across $N$ channels, $\mathbf{W}_k^d \in \mathbb{R}^{D \times D_{ve}}$ and $\mathbf{W}_k^g \in \mathbb{R}^{(D+D_{ve}) \times 1}$ are learnable weights. These hybrid embeddings $\mathbf{E}_{p,1}, \mathbf{E}_{p,2} \in \mathbb{R}^{N \times D_{ve}}$ are then used to calculate the adaptive adjacency matrix $\mathbf{A}_p \in \mathbb{R}^{N \times N}$ for the $p$-th time section, which dynamically captures directional dependencies among channels:

$$\mathbf{A}_p = \mathrm{Softmax}(\mathrm{ReLU}(\mathbf{E}_{p,1}\mathbf{E}_{p,2}^\top)). \qquad (8)$$

Then graph convolution operations with skip connections are applied to exchange information among channels according to $\mathbf{A}_p$:

$$\mathbf{H}_p^{gcn} = \mathrm{ReLU}\left( \sum_{m=0}^M (\mathbf{A}_p)^m \mathbf{H}_p \mathbf{W}_m^{gcn} \right) + \mathbf{H}_p, \quad (9)$$

where $M$ is the number of GCN layers.

Finally, to ensure comprehensive representation while preserving original information, we concatenate the original $\mathbf{H}_p$ with $\mathbf{H}_p^{gcn}$ after cross-channel interaction as inputs $\mathbf{H}_p^{in}$ for MAE, represented as:

$$\mathbf{H}_p^{in} = [\mathbf{H}_p \| \mathbf{H}_p^{gcn}] \in \mathbb{R}^{N \times 2D}, \qquad (10)$$

**2.4. Time-Wise Reconstruction**

After cross-channel complementation, we leverage the capability of visual MAE for modeling semantically sparse multichannel data obtained from pretraining to model temporal dependencies among patches within each channel.

2.4.1. INPUT EMBEDDING AND TEMPORAL POSITION EMBEDDINGS

For a concise representation, if not emphasized in the following, we use $[h_p^{in}]_{p=1}^P \in \mathbb{R}^{P \times 2D}$ to denote the sequence of feature patches within **each single channel**. Before MAE encoding, we compress the cross-channel information complementation and original information using a linear projection $\mathbf{W}_{enc} \in \mathbb{R}^{2D \times D_e}$, to adapt it to the MAE input dimension $D_e$:

$$e_p = h_p^{in} \mathbf{W}_{enc}. \qquad (11)$$

Next, to enable temporal-aware reconstruction, we employ learnable temporal period embeddings, similar to positional embeddings. Specifically, for a sequence of patches of length P, the temporal period embedding for the $p$-th patch is initialized using 2D sine-cosine encoding, represented as:

$$\mathrm{TPE}_p^h[2k] = \sin(p/10000^{2k/d}), \qquad (12)$$

$$\mathrm{TPE}_p^h[2k+1] = \cos(p/10000^{2k/d}), \qquad (13)$$

$$\mathrm{TPE}_p^w[2k] = \sin(1/10000^{2k/d}), \qquad (14)$$

$$\mathrm{TPE}_p^w[2k+1] = \cos(1/10000^{2k/d}), \qquad (15)$$

where $k \in [0, d/4 - 1]$, and $d$ is half of the encoder embedding dimension $D_e/2$ or decoder embedding dimension $D_d/2$. The complete time period embeddings for the $p$-th patch for encoder and decoder are then represented as:

$$\mathrm{TPE}_p^{enc} = [\mathrm{TPE}_p^h[0 : D_e/2] \| \mathrm{TPE}_p^w[D_e/2 : D_e]], \quad (16)$$

$$\text{TPE}_p^{dec} = [\text{TPE}_p^h[0:D_d/2]\|\text{TPE}_p^w[D_d/2:D_d]]. \quad (17)$$

This strategy treats embeddings as a $T \times 1$ patch sequence, allowing the model to adapt MAE's pretrained position understanding capabilities to temporal representations and capture periodic features during optimization. We then add this embedding to $e_p$ to get the inputs of the MAE encoder:

$$e_p^{enc} = e_p + \text{TPE}_p^{enc}. \quad (18)$$

2.4.2. ENCODE AND RECONSTRUCTION

With the input embeddings, MAE aims to learn the temporal dependencies among patches within each channel alongside the cross-channel information complementation, and reconstruct them at target time segments. The embedded sequence is encoded by the MAE encoder $\mathcal{E}$:

$$\{z_p\}_{p=1}^P = \mathcal{E}(\{e_p^{enc}\}_{p=1}^P). \quad (19)$$

For future patch reconstruction, we append $N_{rec}$ learnable mask tokens $\{[M]\}_{i=1}^{N_{rec}}$ to the linearly projected tokens $\{z_p\}_{p=1}^P \boldsymbol{W}_{dec}$. In addition, we concatenate the corresponding temporal positional embeddings $\{\text{TPE}_{P+i}^{dec}\}_{i=1}^{N_{rec}}$ of the target time periods with those of the encoded tokens $\{\text{TPE}_p^{dec}\}_{p=1}^P$. The MAE decoder $\mathcal{D}$ takes this augmented sequence as input:

$$\{\hat{z}_{P+i}^m\}_{i=1}^{N_{rec}} = \mathcal{D}(Z^* + \text{TPE}^*), \quad (20)$$

$$Z^* = [\{z_p\}_{p=1}^P W_{dec}; \{[M]\}_{i=1}^{N_{rec}}], \quad (21)$$

$$\text{TPE}^* = [\{\text{TPE}_p^{dec}\}_{p=1}^P; \{\text{TPE}_{P+i}^{dec}\}_{i=1}^{N_{rec}}], \quad (22)$$

where $[\cdot;\cdot]$ denotes sequence concatenation, $\{\hat{z}_{P+i}^m\}_{i=1}^{N_{rec}}$ represents reconstructed representations of target time periods, and $W_{dec} \in \mathbb{R}^{D_e \times D_d}$ projects the encoded representation into the input dimension of the decoder $\mathbb{R}^{D_d}$.

This approach leverages the visual pre-trained capabilities of MAE, enabling historical and future patch reconstruction during self-supervised training and supervised fine-tuning, respectively.

**2.5. Patch2Point Prediction**

We employ a coarse-to-fine technique to generate predictions for specific timestamps. In the coarse phase, period-level patches are reconstructed via MAE. Then, in the fine-grained phase, these patches are queried with timestamp embeddings for point-level predictions.

In detail, given a query timestamp $t_q$, we first generate a query embedding $\phi(t_q)$ and select its corresponding patch index $i_q$ **matching** $t_{start}^{i_q} \le t_q \le t_{start}^{i_q} + s$, where $s$ is the patch size and stride length.

We calculate the $\text{TPE}_{i_q}^{dec}$ for the target patch and utilize the method introduced in Sec. 2.4.2 to reconstruct the $i_q$-th patch $\hat{z}_{i_q}^m$.

The prediction is generated through a 2-layer MLP network $\mathcal{F}$ that takes the query and the reconstructed patch as input:

$$\hat{x}_q = \mathcal{F}(\phi(t_q), \hat{z}_{i_q}^m). \quad (23)$$

This strategy offers three key advantages: (1) it enables flexible and accurate predictions at arbitrary continuous timestamps within target temporal periods; (2) it comprehensively utilizes patch-level temporal patterns and cross-channel complementary information; (3) it effectively filters out irrelevant information from other temporal-channel contexts, ensuring focused and precise predictions.

**2.6. Training Strategy**

We employ a two-stage strategy for training: self-supervised learning and supervised fine-tuning. The rationale is detailed in Appendix A.3.

**Self-supervised learning for IMTS modeling.** Given a mask ratio $r$, we randomly mask a portion of patches before encoding. Specifically, from the embedded sequence $\{e_p^{enc}\}_{p=1}^P$, we randomly select $|\mathcal{M}| = [r \cdot P]$ patches to mask. The remaining patches $\{e_p^{enc}\}_{p\in\mathcal{V}}$ are encoded:

$$\{z_p\}_{p\in\mathcal{V}} = \mathcal{E}(\{e_p^{enc}\}_{p\in\mathcal{V}}), \quad (24)$$

where $\mathcal{V}$ denotes the set of unmasked indices, $\mathcal{M}$ denotes the set of masked indices. The projected tokens $\{z_p\}_{p\in\mathcal{V}}\boldsymbol{W}_{dec}$ are then added with $\{\text{TPE}_p^{dec}\}_{p\in\mathcal{V}}$ and concatenated with learnable mask tokens $\{[M]\}_{i=1}^{|\mathcal{M}|}$ added with $\{\text{TPE}_p^{dec}\}_{p\in\mathcal{M}}$ to reconstruct $\hat{z}_{i_h}^m$:

$$\{\hat{z}_{i_h}^m\}_{i_h\in\mathcal{M}} = \mathcal{D}(Z^{ssl} + \text{TPE}^{ssl}), \quad (25)$$

$$Z^{ssl} = [\{z_p\}_{p\in\mathcal{V}}W_{dec}; \{[M]\}_{i=1}^{|\mathcal{M}|}], \quad (26)$$

$$\text{TPE}^{ssl} = [\{\text{TPE}_p^{dec}\}_{p\in\mathcal{V}}; \{\text{TPE}_{i_h}^{dec}\}_{i_h\in\mathcal{M}}], \quad (27)$$

The self-supervised training loss is formulated as follows:

$$\mathcal{L}_{ssl} = \frac{1}{N}\sum_{n=1}^N \frac{1}{\mathcal{H}_n}\sum_{h=1}^{\mathcal{H}_n} \|\mathcal{F}(\phi(t_h^n), \hat{z}_{i_h}^{m,n}) - x_h^n\|_2^2, \quad (28)$$

where $\{[t_h^n]_{h=1}^{\mathcal{H}_n}\}_{n=1}^N$ represents the history query timestamps set across $N$ channels with $t_{start}^{i_h} \le t_h^n \le t_{start}^{i_h} + s$. For the $n$-th channel, $\hat{z}_{i_h}^{m,n}$ denotes the reconstructed $i_h$-th patch, and $x_h^n$ is the ground truth value at $t_h^n$.

**Supervised fine-tuning for task adaptation.** We follow the same reconstruction and forecasting process detailed in Sec. 2.4.2 and Sec. 2.5. Here, $\{\text{TPE}_{P+i}^{dec}\}_{i=1}^{N_{rec}}$ represents future TPEs for $N_{rec}$ target periods. Given future query timestamps set $\{[t_q^n]_{q=1}^{\mathcal{Q}_n}\}_{n=1}^N$ across $N$ channels with $t_{start}^{i_q} \le t_q^n \le t_{start}^{i_q} + s$, we minimize the prediction loss:

$$\mathcal{L}_{ft} = \frac{1}{N}\sum_{n=1}^N \frac{1}{\mathcal{Q}_n}\sum_{q=1}^{\mathcal{Q}_n} \|\mathcal{F}(\phi(t_q^n), \hat{z}_{i_q}^{m,n}) - x_q^n\|_2^2, \quad (29)$$

where for the $n$-th channel, $\hat{z}_{i_q}^{m,n}$ denotes the reconstructed $i_q$-th patch, and $x_q^n$ is the ground truth value at $t_q^n$. During this stage, we selectively optimize some components to adapt to the forecasting task while preserving basic capabilities, which is detailed in Sec. 3.1.

*Table 1.* Overall performance evaluated by MAE and MSE ($mean \pm std$). The best-performing results are highlighted in **bold**, the second-best results are highlighted in **blue bold**, and the third-best results are highlighted in underline. 'Zero' and 'Linear' are different imputation methods adapting IMTS to VisionTS. '*' denotes that the performance are reproduced following the original paper.

| Algorithms | PhysioNet | | Human Activity | | USHCN | | MIMIC | |
|---|---|---|---|---|---|---|---|---|
| | MSE×$10^{-3}$ | MAE×$10^{-2}$ | MSE×$10^{-3}$ | MAE×$10^{-2}$ | MSE×$10^{-3}$ | MAE×$10^{-2}$ | MSE×$10^{-2}$ | MAE×$10^{-2}$ |
| DLinear | $41.86 \pm 0.05$ | $15.52 \pm 0.03$ | $4.03 \pm 0.01$ | $4.21 \pm 0.01$ | $6.21 \pm 0.00$ | $3.88 \pm 0.02$ | $4.90 \pm 0.00$ | $16.29 \pm 0.05$ |
| TimesNet | $16.48 \pm 0.11$ | $6.14 \pm 0.03$ | $3.12 \pm 0.01$ | $3.56 \pm 0.02$ | $5.58 \pm 0.05$ | $3.60 \pm 0.04$ | $5.88 \pm 0.08$ | $13.62 \pm 0.07$ |
| PatchTST | $12.00 \pm 0.23$ | $6.02 \pm 0.14$ | $4.29 \pm 0.14$ | $4.80 \pm 0.09$ | $5.75 \pm 0.01$ | $3.57 \pm 0.02$ | $3.78 \pm 0.03$ | $12.43 \pm 0.10$ |
| Crossformer | $6.66 \pm 0.11$ | $4.81 \pm 0.11$ | $4.29 \pm 0.20$ | $4.89 \pm 0.17$ | $5.25 \pm 0.04$ | $3.27 \pm 0.09$ | $2.65 \pm 0.10$ | $9.56 \pm 0.29$ |
| Graph Wavenet | $6.04 \pm 0.28$ | $4.41 \pm 0.11$ | $2.89 \pm 0.03$ | $3.40 \pm 0.05$ | $5.29 \pm 0.04$ | $3.16 \pm 0.09$ | $2.93 \pm 0.09$ | $10.50 \pm 0.15$ |
| MTGNN | $6.26 \pm 0.18$ | $4.46 \pm 0.07$ | $3.03 \pm 0.03$ | $3.53 \pm 0.03$ | $5.39 \pm 0.05$ | $3.34 \pm 0.02$ | $2.71 \pm 0.23$ | $9.55 \pm 0.65$ |
| StemGNN | $6.86 \pm 0.28$ | $4.76 \pm 0.19$ | $8.81 \pm 0.37$ | $6.90 \pm 0.02$ | $5.75 \pm 0.09$ | $3.40 \pm 0.09$ | $1.73 \pm 0.02$ | $7.71 \pm 0.11$ |
| CrossGNN | $7.22 \pm 0.36$ | $4.96 \pm 0.12$ | $3.03 \pm 0.10$ | $3.48 \pm 0.08$ | $5.66 \pm 0.04$ | $3.53 \pm 0.05$ | $2.95 \pm 0.16$ | $10.82 \pm 0.21$ |
| FourierGNN | $6.84 \pm 0.35$ | $4.65 \pm 0.12$ | $2.99 \pm 0.02$ | $3.42 \pm 0.02$ | $5.82 \pm 0.06$ | $3.62 \pm 0.07$ | $2.55 \pm 0.03$ | $10.22 \pm 0.08$ |
| GRU-D | $5.59 \pm 0.09$ | $4.08 \pm 0.05$ | $2.94 \pm 0.05$ | $3.53 \pm 0.06$ | $5.54 \pm 0.38$ | $3.40 \pm 0.28$ | $1.76 \pm 0.03$ | $7.53 \pm 0.09$ |
| SeFT | $9.22 \pm 0.18$ | $5.40 \pm 0.08$ | $12.20 \pm 0.17$ | $8.43 \pm 0.07$ | $5.80 \pm 0.19$ | $3.70 \pm 0.11$ | $1.87 \pm 0.01$ | $7.84 \pm 0.08$ |
| RainDrop | $9.82 \pm 0.08$ | $5.57 \pm 0.06$ | $14.92 \pm 0.14$ | $9.45 \pm 0.05$ | $5.78 \pm 0.22$ | $3.67 \pm 0.17$ | $1.99 \pm 0.03$ | $8.27 \pm 0.07$ |
| Warpformer | $5.94 \pm 0.35$ | $4.21 \pm 0.12$ | $2.79 \pm 0.04$ | $3.39 \pm 0.03$ | $5.25 \pm 0.05$ | $3.23 \pm 0.05$ | $1.73 \pm 0.04$ | $7.58 \pm 0.13$ |
| mTAND | $6.23 \pm 0.24$ | $4.51 \pm 0.17$ | $3.22 \pm 0.07$ | $3.81 \pm 0.07$ | $5.33 \pm 0.05$ | $3.26 \pm 0.10$ | $1.85 \pm 0.06$ | $7.73 \pm 0.13$ |
| Latent-ODE | $6.05 \pm 0.57$ | $4.23 \pm 0.26$ | $3.34 \pm 0.11$ | $3.94 \pm 0.12$ | $5.62 \pm 0.03$ | $3.60 \pm 0.12$ | $1.89 \pm 0.19$ | $8.11 \pm 0.52$ |
| CRU | $8.56 \pm 0.26$ | $5.16 \pm 0.09$ | $6.97 \pm 0.78$ | $6.30 \pm 0.47$ | $6.09 \pm 0.17$ | $3.54 \pm 0.18$ | $1.89 \pm 0.19$ | $8.11 \pm 0.52$ |
| Neural Flow | $7.20 \pm 0.07$ | $4.67 \pm 0.04$ | $4.05 \pm 0.13$ | $4.46 \pm 0.09$ | $5.35 \pm 0.05$ | $3.25 \pm 0.05$ | $1.87 \pm 0.05$ | $8.03 \pm 0.19$ |
| T-PATCHGNN | $4.98 \pm 0.08$ | $3.72 \pm 0.03$ | $\mathbf{2.66 \pm 0.03}$ | $3.15 \pm 0.02$ | $\mathbf{5.00 \pm 0.04}$ | $3.08 \pm 0.04$ | $\mathbf{1.36 \pm 0.02^*}$ | $6.56 \pm 0.11^*$ |
| VisionTS (zero) | $42.41 \pm 0.02$ | $13.13 \pm 0.02$ | $8.45 \pm 0.01$ | $6.55 \pm 0.04$ | $7.89 \pm 0.05$ | $5.13 \pm 0.05$ | $8.61 \pm 0.00$ | $18.76 \pm 0.01$ |
| VisionTS (linear) | $40.50 \pm 0.05$ | $12.85 \pm 0.01$ | $7.77 \pm 0.01$ | $6.27 \pm 0.04$ | $6.77 \pm 0.04$ | $4.32 \pm 0.03$ | $7.74 \pm 0.02$ | $17.06 \pm 0.05$ |
| VIMTS(20% data) | $4.93 \pm 0.09$ | $3.63 \pm 0.06$ | $2.72 \pm 0.01$ | $3.14 \pm 0.00$ | $5.01 \pm 0.01$ | $3.06 \pm 0.04$ | $1.47 \pm 0.01$ | $6.71 \pm 0.06$ |
| VIMTS(50% data) | $\mathbf{4.86 \pm 0.09}$ | $\mathbf{3.57 \pm 0.05}$ | $2.69 \pm 0.01$ | $\mathbf{3.11 \pm 0.01}$ | $4.86 \pm 0.01$ | $\mathbf{2.97 \pm 0.08}$ | $\mathbf{1.41 \pm 0.01}$ | $\mathbf{6.47 \pm 0.12}$ |
| VIMTS(100% data) | $\mathbf{4.81 \pm 0.07}$ | $\mathbf{3.54 \pm 0.04}$ | $\mathbf{2.65 \pm 0.01}$ | $\mathbf{3.08 \pm 0.01}$ | $4.86 \pm 0.02$ | $\mathbf{2.98 \pm 0.05}$ | $\mathbf{1.36 \pm 0.02}$ | $\mathbf{6.40 \pm 0.17}$ |

*Table 2.* Datasets Technical Specifications

| Description | PhysioNet | Human Activity | USHCN | MIMIC |
|---|---|---|---|---|
| Samples | 12,000 | 5,400 | 1,114 | 23,457 |
| Channels | 41 | 12 | 5 | 96 |
| Missing ratio | 85.7% | 75.0% | 77.9% | 96.7% |
| Observation | 24 h | 3,000 ms | 24 months | 24 h |
| Prediction | Next 24 h | Next 1,000 ms | Next month | Next 24 h |

# 3. Experiments

## 3.1. Experimental Setup

**Datasets and Evaluation Metrics.** To evaluate the performance of models on the IMTS forecasting task, we utilize four datasets from diverse domains: healthcare (**PhysioNet** (Silva et al., 2012), **MIMIC**(Johnson et al., 2016)), biomechanics (**Human Activity**), and climate science (**USHCN** (Menne et al., 2015)). Technical specifications for these datasets are summarized in Table 2. Each dataset is split into training, validation, and test sets at ratios of 60%, 20%, and 20%, respectively. Performance is measured using Mean Squared Error (MSE) and Mean Absolute Error (MAE), where MAE $= \frac{1}{|\mathcal{Q}|} \sum_{i=1}^{|\mathcal{Q}|} |x_i - \hat{x}_i|$ and MSE $= \frac{1}{|\mathcal{Q}|} \sum_{i=1}^{|\mathcal{Q}|} (x_i - \hat{x}_i)^2$, with $x_i$, $\hat{x}_i$, and $|\mathcal{Q}|$ representing the ground truth, predicted value, and the number of queries, respectively.

**Implementation Details.** Experiments are performed on individual NVIDIA RTX 4090 GPUs. For VIMTS setups, we set the hidden dimensions to 32 for USHCN and PhysioNet, 40 for MIMIC, and 64 for Human Activity. The batch size is 32 for the two training stages of PhysioNet and Human Activity's pre-training stage, 64 for the fine-tuning stage of Human Activity and the two training stages of USHCN, 12 for MIMIC's pre-training stage, and 16 for

its fine-tuning stage. We use visual MAE-base (He et al., 2022) as the backbone, the Adam optimizer with a learning rate of $1 \times 10^{-4}$ for training, and apply early stopping if the validation loss does not decrease for 15 consecutive epochs. To ensure robustness, each experiment is repeated with five different random seeds, and the mean and standard deviation of the results are reported. Further hyperparameter details are elaborated in Appendix A.5.

**Parameter Optimization Details.** We optimize all parameters during the self-supervised learning stage. In the fine-tuning stage, for USHCN, PhysioNet, and Human Activity, we freeze the GCN and MAE (except for normalization layers); for MIMIC, we only freeze the MAE (except for normalization layers, position embedding, and patch projection layer).

**Baselines.** To establish a comprehensive benchmark for the IMTS forecasting task, we select baselines from four methodological domains. Specifically, we include: (1) **MTS Forecasting**: DLinear (Zeng et al., 2023), TimesNet (Wu et al., 2022), PatchTST (Nie et al., 2022), Crossformer (Zhang et al., 2022), GraphWaveNet (Wu et al., 2019), MT-GNN (Wu et al., 2020), StemGNN (Cao et al., 2020), Cross-GNN (Huang et al., 2023), FourierGNN (Yi et al., 2024) and VisionTS (Chen et al., 2025); (2) **IMTS Classification**: GRU-D (Che et al., 2018), SeFT (Horn et al., 2020), RainDrop (Zhang et al., 2022), Warpformer (Zhang et al., 2023); (3) **IMTS Interpolation**: mTAND (Shukla & Marlin, 2021a); (4) **IMTS Forecasting**: Latent ODEs (Rubanova et al., 2019), CRU (Schirmer et al., 2022), Neural Flows (Biloš et al., 2021), and t-PatchGNN (Zhang et al., 2024a).

*Table 3.* Ablation results of VIMTS on four datasets evaluated by MAE and MSE (mean ± std). The best-performing results are highlighted in **bold**

| Ablation | PhysioNet | | Human Activity | | USHCN | | MIMIC | |
|---|---|---|---|---|---|---|---|---|
| | MSE×$10^{-3}$ | MAE×$10^{-2}$ | MSE×$10^{-3}$ | MAE×$10^{-2}$ | MSE×$10^{-1}$ | MAE×$10^{-1}$ | MSE×$10^{-2}$ | MAE×$10^{-2}$ |
| Complete | **4.81 ± 0.07** | **3.54 ± 0.04** | **2.65 ± 0.01** | **3.08 ± 0.01** | **4.86 ± 0.02** | 2.98 ± 0.05 | **1.36 ± 0.02** | **6.40 ± 0.17** |
| w/o Pre | 5.13 ± 0.04 | 3.75 ± 0.04 | 2.73 ± 0.02 | 3.16 ± 0.02 | 4.95 ± 0.01 | 3.05 ± 0.08 | 1.39 ± 0.02 | 6.49 ± 0.08 |
| w/o SSL | 5.46 ± 0.30 | 3.93 ± 0.28 | 2.76 ± 0.08 | 3.26 ± 0.11 | 5.05 ± 0.06 | 3.14 ± 0.14 | 1.41 ± 0.03 | 6.67 ± 0.15 |
| w/o Pre & SSL | 5.70 ± 0.42 | 4.24 ± 0.33 | 2.84 ± 0.06 | 3.32 ± 0.09 | 5.04 ± 0.04 | 3.06 ± 0.06 | 1.45 ± 0.05 | 6.99 ± 0.33 |
| w/o GCN | 4.94 ± 0.03 | 3.55 ± 0.03 | 2.66 ± 0.01 | 3.08 ± 0.01 | 4.93 ± 0.01 | **2.97 ± 0.07** | 2.25 ± 0.02 | 8.82 ± 0.15 |
| rp Transformer | 5.57 ± 0.34 | 3.99 ± 0.24 | 2.84 ± 0.07 | 3.32 ± 0.10 | 5.09 ± 0.06 | 3.14 ± 0.13 | 1.40 ± 0.04 | 6.66 ± 0.14 |

*Table 4.* Patch2Point vs. Direct Projection of VIMTS on four datasets evaluated by MAE and MSE (mean ± std). In detail, training stages marked with ✓ applied with Patch2Point, the others represent stages with direct projection heads. The best/worst-performing results are highlighted in **bold**/**red bold**.

| Patch2Point | | PhysioNet | | Human Activity | | USHCN | | MIMIC | |
|---|---|---|---|---|---|---|---|---|---|
| SSL | Finetune | MSE×$10^{-3}$ | MAE×$10^{-2}$ | MSE×$10^{-3}$ | MAE×$10^{-2}$ | MSE×$10^{-1}$ | MAE×$10^{-1}$ | MSE×$10^{-2}$ | MAE×$10^{-2}$ |
| ✓ | ✓ | **4.81 ± 0.07** | **3.54 ± 0.04** | **2.65 ± 0.01** | **3.08 ± 0.01** | **4.86 ± 0.02** | **2.98 ± 0.05** | 1.36 ± 0.02 | 6.40 ± 0.17 |
| ✓ | | 4.97 ± 0.07 | 3.62 ± 0.05 | 2.72 ± 0.01 | 3.17 ± 0.02 | **5.03 ± 0.02** | 3.09 ± 0.13 | **1.34 ± 0.01** | **6.37 ± 0.08** |
| | | **4.98 ± 0.07** | **3.62 ± 0.03** | **2.74 ± 0.01** | **3.20 ± 0.01** | 5.01 ± 0.04 | **3.10 ± 0.10** | **1.38 ± 0.02** | **6.50 ± 0.15** |

This selection ensures cross-methodological comparisons across regular/irregular time series forecasting, classification, and interpolation tasks, providing a robust evaluation of generalization capabilities.

## 3.2. Main results

We evaluated VIMTS against 20 baseline models across different domains: clinical (PhysioNet, MIMIC), biomechanics (Human Activity), and climate (USHCN), using MSE and MAE as performance metrics, as shown in Table 1. Conventional methods like DLinear, TimesNet, and PatchTST, while effective for regular time series, struggle with IMTS due to their inability to handle irregular sampling and cross-channel dependencies, leading to significant errors. VisionTS, whether using Zero' or Linear' interpolation for data adaptation, fails to perform well on IMTS tasks. This highlights the inadequacy of existing vision foundation model-based methods in dealing with the missing values, varying data structures, and complex temporal and cross-channel dependencies of IMTS data.

In contrast, VIMTS consistently outperforms other methods, including the currently best-performing baseline, t-PatchGNN. When only 20% or 50% of training data are available, VIMTS matches the performance of t-PatchGNN with complete data and exceeds the performance of all other methods. Increasing the utilization of training data further to 100%, VIMTS demonstrates even better performance, achieving the lowest MSE and MAE across all four real-world datasets. These results validate VIMTS's superior adaptability and effectiveness in handling IMTS forecasting tasks.

## 3.3. Ablation Study

To validate the necessity of core components in VIMTS, we conducted an ablation study comparing multiple variants. (1) **Complete** represents the model without any ablation; (2)

**w/o Pre** removes the visual pre-training of MAE; (3) **w/o SSL** skips IMTS-specific self-supervised training; (4) **w/o Pre & SSL** trains the model entirely from scratch without using visual pre-training or self-supervised learning; (5) **w/o GCN** removes cross-channel graph convolutions; (6) **rp Transformer** replaces MAE with a vanilla Transformer encoder. In addition, we also explore the effectiveness of the Patch2Point prediction by replacing the coarse-to-fine predictor with a flatten-projection layer in self-supervised learning stage and fine-tuning stage.

As shown in Table 3 and Table 4, replacing MAE with a standard Transformer leads to a significant performance drop, demonstrating MAE's architectural advantage for handling semantically sparse data across multiple channels. Similarly, removing either visual pre-training (w/o Pre) or self-supervised learning (w/o SSL) results in notable performance declines, highlighting that visual priors provide valuable initialization while SSL helps adapt to IMTS-specific characteristics. Moreover, the ablation of GCN and coarse-to-fine decoding show their effectiveness: while individual channels may suffer from missing values, GCN compensates through cross-channel dependencies. Additionally, ablation studies on the Patch2Point predictor's two-stage training demonstrate its coarse-to-fine strategy enhances forecasting precision. It achieves this by focusing on relevant temporal context and reducing interference from unrelated time periods, thereby outperforming direct timestamp prediction which is susceptible to irrelevant information.

## 3.4. Few-Shot Learning

To evaluate the few-shot capability of our method and the contributions of visual pre-training (Pre) and self-supervised learning (SSL), we conduct experiments on different few-shot scenarios (10%, 20%, and 50% of the entire training data) and ablation settings (complete, w/o Pre, w/o SSL and w/o Pre & SSL). T-PatchGNN serves as a baseline model.

*Table 5.* Few-shot results of VIMTS on four datasets evaluated by MAE and MSE (mean ± std). The best-performing results are highlighted in **bold**, the second-best results are highlighted in **blue bold**.

| Model | | VIMTS | | t-PatchGNN | | VIMTS(w/o Pre & SSL) | | VIMTS(w/o SSL) | | VIMTS(w/o Pre) | |
|---|---|---|---|---|---|---|---|---|---|---|---|
| Dataset | Ratio | $MSE \times 10^{-3}$ | $MAE \times 10^{-2}$ | $MSE \times 10^{-3}$ | $MAE \times 10^{-2}$ | $MSE \times 10^{-3}$ | $MAE \times 10^{-2}$ | $MSE \times 10^{-3}$ | $MAE \times 10^{-2}$ | $MSE \times 10^{-3}$ | $MAE \times 10^{-2}$ |
| Activity | 0.1 | **2.87 ± 0.02** | **3.24 ± 0.03** | **3.21 ± 0.07** | **3.62 ± 0.10** | 7.28 ± 5.72 | 5.81 ± 2.95 | 3.31 ± 0.14 | 3.65 ± 0.10 | 13.15 ± 5.07 | 8.81 ± 2.35 |
| | 0.2 | **2.72 ± 0.01** | **3.14 ± 0.00** | 3.01 ± 0.06 | **3.42 ± 0.08** | 4.24 ± 0.22 | 3.45 ± 0.14 | **3.00 ± 0.11** | 3.43 ± 0.13 | 4.25 ± 0.06 | 4.14 ± 0.03 |
| | 0.5 | **2.69 ± 0.01** | **3.11 ± 0.01** | **2.90 ± 0.10** | 3.35 ± 0.06 | 2.91 ± 0.10 | 3.36 ± 0.14 | 2.91 ± 0.08 | 3.35 ± 0.12 | 2.92 ± 0.06 | **3.31 ± 0.06** |
| Dataset | Ratio | $MSE \times 10^{-3}$ | $MAE \times 10^{-2}$ | $MSE \times 10^{-3}$ | $MAE \times 10^{-2}$ | $MSE \times 10^{-3}$ | $MAE \times 10^{-2}$ | $MSE \times 10^{-3}$ | $MAE \times 10^{-2}$ | $MSE \times 10^{-3}$ | $MAE \times 10^{-2}$ |
| PhysioNet | 0.1 | **5.16 ± 0.26** | **3.73 ± 0.17** | 7.09 ± 0.24 | 4.21 ± 0.18 | 6.03 ± 0.45 | 4.25 ± 0.17 | 6.35 ± 0.71 | 4.46 ± 0.42 | **5.93 ± 0.06** | **4.17 ± 0.06** |
| | 0.2 | **4.92 ± 0.09** | **3.63 ± 0.06** | 7.23 ± 0.22 | 4.24 ± 0.22 | 5.87 ± 0.47 | 4.21 ± 0.23 | 6.04 ± 0.67 | 4.26 ± 0.37 | **5.45 ± 0.08** | **3.89 ± 0.09** |
| | 0.5 | **4.86 ± 0.09** | **3.57 ± 0.05** | 6.97 ± 0.13 | 4.03 ± 0.13 | 5.60 ± 0.45 | 4.09 ± 0.31 | 5.80 ± 0.40 | 4.07 ± 0.25 | **5.29 ± 0.07** | **3.8 ± 0.02** |
| Dataset | Ratio | $MSE \times 10^{-1}$ | $MAE \times 10^{-1}$ | $MSE \times 10^{-1}$ | $MAE \times 10^{-1}$ | $MSE \times 10^{-1}$ | $MAE \times 10^{-1}$ | $MSE \times 10^{-1}$ | $MAE \times 10^{-1}$ | $MSE \times 10^{-1}$ | $MAE \times 10^{-1}$ |
| USHCN | 0.1 | **5.09 ± 0.07** | **3.12 ± 0.04** | 5.19 ± 0.02 | 3.26 ± 0.06 | 5.23 ± 0.05 | 3.19 ± 0.15 | 5.32 ± 0.08 | 3.28 ± 0.11 | **5.12 ± 0.01** | **3.13 ± 0.05** |
| | 0.2 | **5.01 ± 0.02** | **3.06 ± 0.04** | 5.11 ± 0.05 | 3.18 ± 0.07 | 5.26 ± 0.13 | 3.18 ± 0.07 | 5.24 ± 0.04 | 3.16 ± 0.04 | **5.05 ± 0.02** | **3.07 ± 0.05** |
| | 0.5 | **4.86 ± 0.01** | **2.97 ± 0.08** | 5.04 ± 0.03 | **3.07 ± 0.07** | 5.10 ± 0.02 | 3.16 ± 0.14 | 5.10 ± 0.04 | 3.15 ± 0.08 | **4.99 ± 0.02** | 3.13 ± 0.09 |
| Dataset | Ratio | $MSE \times 10^{-2}$ | $MAE \times 10^{-2}$ | $MSE \times 10^{-2}$ | $MAE \times 10^{-2}$ | $MSE \times 10^{-2}$ | $MAE \times 10^{-2}$ | $MSE \times 10^{-2}$ | $MAE \times 10^{-2}$ | $MSE \times 10^{-2}$ | $MAE \times 10^{-2}$ |
| MIMIC | 0.1 | **1.53 ± 0.01** | **7.06 ± 0.10** | 1.67 ± 0.16 | 7.66 ± 0.55 | 1.71 ± 0.12 | 7.85 ± 0.42 | 1.77 ± 0.13 | 8.23 ± 0.45 | **1.61 ± 0.01** | **7.18 ± 0.20** |
| | 0.2 | **1.47 ± 0.01** | **6.71 ± 0.06** | **1.46 ± 0.05** | 7.07 ± 0.32 | 1.64 ± 0.10 | 7.62 ± 0.35 | 1.57 ± 0.07 | 7.30 ± 0.29 | 1.53 ± 0.03 | **6.90 ± 0.05** |
| | 0.5 | **1.41 ± 0.01** | **6.47 ± 0.12** | **1.39 ± 0.02** | 6.70 ± 0.13 | 1.46 ± 0.03 | 6.93 ± 0.19 | 1.45 ± 0.03 | 6.92 ± 0.14 | 1.43 ± 0.00 | **6.64 ± 0.08** |

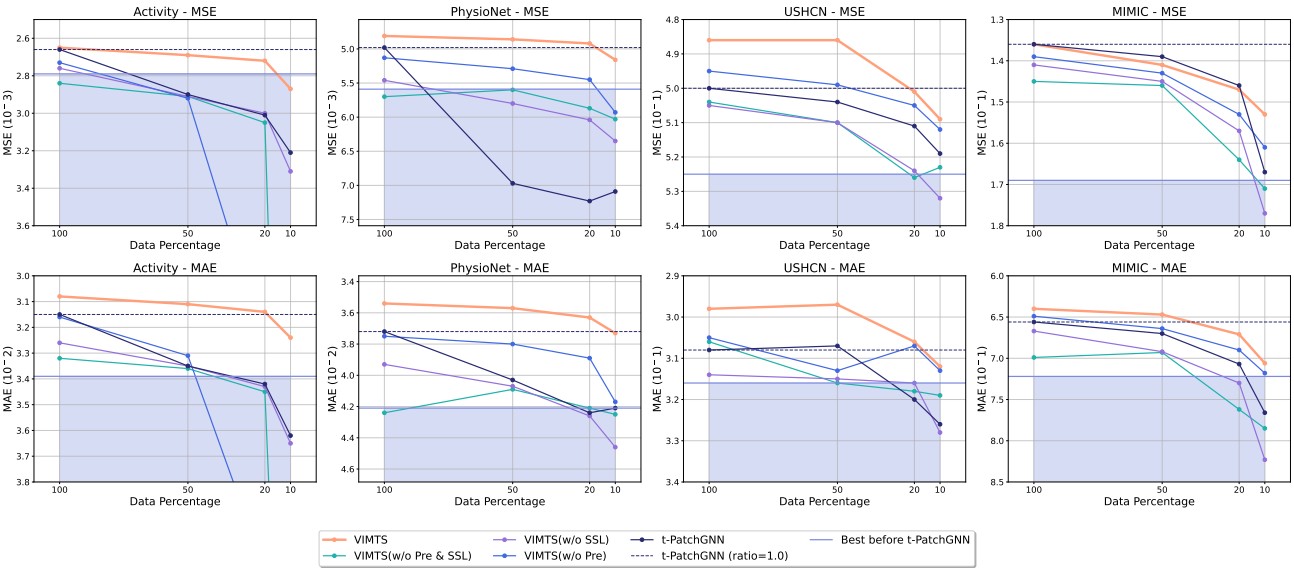

*Figure 4.* Performance comparison of VIMTS model variants under low-resource conditions (10%, 20%, 50%, and 100% data) on MSE and MAE metrics across Activity, PhysioNet, USHCN, and MIMIC datasets. Variants include models with and without visual pre-trained initialization and self-supervised learning. T-PatchGNN is used as a baseline. Lower values indicate better performance.

As shown in Figure 4, VIMTS achieves superior performance across few-shot settings, with lower prediction errors and more stable response curves under varying training data availability. While t-PatchGNN exhibits sensitivity to training data scarcity, VIMTS demonstrates more substantial few-shot generalization, which may benefit from integrating visual pre-training and self-supervised learning. To confirm this, we conducted ablation studies and found that removing either component may degrade performance stability. Although trends differ slightly across the four datasets, these results collectively highlight that the synergistic integration of cross-domain visual pre-training and self-supervised learning enables effective key pattern extraction from limited data, thus advancing sample-efficient IMTS modeling.

## 4. Analysis of Computational Cost

### 4.1. Time and Space Complexity Comparison

**Parameter Numbers.** As shown in Table 6, VIMTS utilizes visual MAE-base as its backbone. This backbone can be fine-tuned on a single NVIDIA RTX 4090. Furthermore, the GCN layers in VIMTS are lightweight, contributing only around 32.4k parameters. The number of trainable parameters is acceptable in real-world applications and lower than other vision foundation model-based methods, such as ViTST.

**Time Complexity.** As shown in Table 6, VIMTS has acceptable training efficiency (87.8 s/epoch) while its inference speed (4.815 ms/instance) outperforms ViTST (visual

*Table 6.* The Temporal and Spacial Complexity of Different Methods.

| Methods | VIMTS | VIMTS w/o GCN | ViTST | tPatchGNN | CRU | WrapFormer |
|---|---|---|---|---|---|---|
| **Training Time (s/epoch)** | 87.80 | 84.96 | 140.2 | 17.14 | 172.45 | 22.67 |
| **Inference Time (ms/sample)** | 4.82 | 4.34 | 7.09 | 1.60 | 7.66 | 5.48 |
| **Trainable Param** | 111.01M | 110.98M | 212.31M | 943.99k | 28.65k | 178.06K |

foundation model, 7.094 ms/instance), CRU (Neural-ODE, 7.655 ms/instance), and WrapFormer (Transformer, 5.475 ms/instance), making it practical for real-world deployment. Though its inference is slightly slower than lightweight models like t-PatchGNN (1.604 ms/instance), it's still efficient and practical in most accuracy-first application scenarios, with a sub-5 ms latency.

### 4.2. The Trade-off between Self-Supervised Learning and Computational Cost

Self-Supervised Learning (SSL) effectively offers a favorable trade-off between computational cost and performance, which is evident in the following three aspects.

**High Data Efficiency and Few-Shot Capability**. With SSL, VIMTS achieves competitive results against state-of-the-art models on four IMTS datasets while utilizing only 20% of the training data. This significant reduction of required data volume accelerates model development and deployment, especially in data-scarce environments.

**Manageable Model Complexity**. Our analysis demonstrates excellent performance without requiring excessive scaling. VIMTS typically uses approximately three lightweight GCN layers and several simple MLP predictors, aside from the MAE backbone. Previous trials show that scaling MAE beyond its base level doesn't improve results and often leads to memory issues, consistent with observations in VisionTS (Chen et al., 2025), another model that applies visual MAE to RTS forecasting.

**Efficient Inference and Improved Performance**. While Self-Supervised Learning (SSL) increases overall training time by approximately 40%, the training time per epoch is still faster compared to CRU and ViTST, which remains acceptable. Importantly, SSL doesn't increase inference cost, allowing VIMTS to maintain faster inference speeds than most competitors while remaining competitive with t-PatchGNN. Given that real-world applications often face data limitations and prioritize inference efficiency over training expenses, this trade-off, which involves accepting a reasonable increase in training time for improved accuracy and efficient inference, offers significant practical advantages.

### 5. Conclusion

This paper introduced VIMTS, a pioneering framework that leverages the capability of visual pre-trained MAE for modeling semantically sparse multichannel data for IMTS forecasting. To mitigate the effect of missing values, VIMTS processes sparse IMTS along the timeline into image-like patches with equal-intervals, then complements these patches with information from related channels using learned cross-channel dependencies. Then it leverages the capability of visual MAE for handling sparse multichannel data for patch reconstruction, followed by a coarse-to-fine technique that progressively generates precise predictions from focused context. The framework is trained with a two-stage strategy. First, self-supervised learning is employed to enhance IMTS data modeling by adapting visual MAE's strengths to IMTS data, while supervised fine-tuning is applied as follows for task-specific adaptation. Extensive experiments on four real-world datasets demonstrate VIMTS' superior performance and robust few-shot capabilities, achieving competitive accuracy even with limited data compared to baselines trained on full datasets, paving the way for applying visual foundation models to more general time series forecasting tasks.

### 6. Limitations and Future Work

While VIMTS advances IMTS forecasting, limitations persist in scalability and structural flexibility. For scalability, the design of a larger IMTS foundation model to achieve more powerful performance and better generalization across downstream datasets remains a problem, which may be resolved by constructing larger-scale pre-training datasets and developing well-designed fine-tuning strategies. For structural flexibility, current models are limited to fixed patch sizes and the number of channels, struggling with dynamic data structures, thereby hindering true zero-shot capabilities without parameter tuning. Future directions should prioritize 'time-contextual scaling' mechanisms that dynamically adjust semantic hierarchies using timestamp metadata and a general cross-channel dependency graph foundation model that flexibly handles information exchange among any number of channels.

## Impact Statement

This paper aims to advance research in Irregular Multivariate Time Series (IMTS) prediction. There exist many potential societal consequences of our work, but none that we feel require specific highlighting here.

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

# A. Appendix

## A.1. Related Work

### A.1.1. Irregular Multivariate Time Series Forecasting

Irregularly sampled time series forecasting poses unique challenges due to non-uniform observation intervals across multiple variables (Shukla & Marlin, 2021b; Horn et al., 2020). Basic approaches include fixed discretization, which simplifies processing but introduces missing data issues (Marlin et al., 2012; Lipton et al., 2016), and interpolation methods that improve robustness by leveraging past and future data (Yoon et al., 2018; Horn et al., 2020). Recent advancements have been made with Neural ODEs (Chen et al., 2018) for continuous dynamics modeling, extended by frameworks like ODE-RNN (Rubanova et al., 2019) and Neural CDE (Kidger et al., 2020), enhancing efficiency and adaptability. CRU (Schirmer et al., 2022) integrates probabilistic models for better interval management, while GNN-based approaches, such as RAINDROP (Zhang et al., 2022), model data point interactions effectively for global understanding.

### A.1.2. Foundation Model for Time Series Forecasting

Foundation models (Bommasani et al., 2022) have transformed time series forecasting, notably through the adaptation of masked autoencoders (MAEs) originally designed for visual tasks (He et al., 2022). VisionTS (Chen et al., 2025) innovates by treating time series forecasting as image reconstruction, achieving zero-shot performance through MAE's patch-level reconstruction paradigm. This method captures temporal patterns without modality-specific adaptations, outperforming language-model-based approaches like Time-LLM (Jin et al., 2024).

## A.2. Methodology Comparison and Clarification

### A.2.1. Clarification of Originality and differentials from (Jungo et al., 2024).

While both works explore patchification, our core innovation lies in the exploration of visual MAE's architecture and the capability benefiting from visual initialization and self-supervised learning for unstructured IMTS data reconstruction. Beyond this, VIMTS distinguishes itself through several key aspects not addressed by (Jungo et al., 2024).

**Enhanced Channel Dependency Modeling with GCN**. Unlike Jungo's projection-based forecasting, VIMTS leverages Graph Convolutional Networks (GCNs) to capture both static and dynamic information. This allows it to model the bidirectional information flow among channels, which is more consistent with real-world information dependencies and enables explicit cross-channel compensation to effectively impute missing values. Jungo et al.'s work (Jungo et al., 2024) solely relies on projection, lacking this sophisticated cross-channel dependency modeling. Our ablation study confirms the significant performance boost from our GCN module.

**Leveraging Vision Pretraining**. A crucial aspect of VIMTS is our explicit utilization and exploration of vision pretraining's foundational capabilities and their importance for the model's overall performance. This vital component, which provides a strong initialization for sparse pattern learning and few-shot learning, is not mentioned or explored in (Jungo et al., 2024).

**Fine-Grained Time × Channel Prediction**. VIMTS employs a coarse-to-fine prediction strategy, allowing for more precise predictions at specific time segments and channels. In contrast, Jungo's projection-based approach operates at a coarser level. We have even conducted experiments with an encoder architecture using direct projection at different training stages, similar to (Jungo et al., 2024), which yielded inferior results, further highlighting the advantage of the proposed VIMTS.

### A.2.2. Comparison with Other Mask-Reconstruction Methods

Our experiments include state-of-the-art masking-based transformer variants such as PatchTST and VisionTS, and demonstrate VIMTS' superior performance, which benefits from its ability to effectively handle IMTS irregularities and missingness. While the transformer-based models with mask reconstruction, PatchTST (Nie et al., 2022) have shown strong performance on regularly sampled Multivariate Time Series (MTS) data, they lack explicit mechanisms to handle irregular sampling prevalent in IMTS data. Their design also does not incorporate cross-channel modeling, which is crucial for imputation in the presence of missingness, leading to significant performance drops. Moreover, VisionTS (Chen et al., 2025), a vision-based foundation model with masked reconstruction, performs well on MTS but fails to generalize to IMTS tasks due to its reliance on rigid grid-like patch resizing and normalization, which may cause information loss. Similarly, MOMENT (Goswami et al., 2024), another transformer-based foundation model with mask-reconstruction underperforms even PatchTST on MTS

tasks and struggles to address irregular sampling or missing value issues.

### A.2.3. COMPARISON WITH ViTST (LI ET AL., 2023)

Although both ViTST and VIMTS employ image-like representations for time series, there are key distinctions in Innovation Points: To extract information from IMTS data, ViTST deals with missingness with interpolation and by transforming IMTS to images, which causes information loss with additional computational overhead, makes inputs less precise for understanding patterns in data, and fails to perform well in forecasting. In contrast, our model divides data by time intervals, patchifies it into time × channel patches, and extracts block features without interpolation, which preserves data integrity and creates more precise inputs for MAE to model internal data structures without computational overhead from image construction. Further, it explicitly models cross-channel interaction with GCN, thereby compensating for missingness across channels. Our analysis is further supported by empirical evidence, which evaluates the model performance on PhysioNet.

| Method | MSE($\times 10^{-3}$) | MAE($\times 10^{-2}$) |
|---|---|---|
| VIMTS (Ours) | $4.81 \pm 0.07$ | $3.54 \pm 0.04$ |
| ViTST | $66.37 \pm 0.08$ | $20.16 \pm 0.05$ |
| VisionTS (Zero Interp.) | $42.41 \pm 0.02$ | $13.13 \pm 0.02$ |
| VisionTS (Linear Interp.) | $40.50 \pm 0.05$ | $12.85 \pm 0.01$ |

*Table 7.* Comparison of different methods.

Note that this experiment also involves VisionTS (Chen et al., 2025) with similar image-based methods, confirming that despite its powerful visual MAE framework and strong performance on regularly sampled time series, it similarly deteriorates when processing irregular data.

As for computational cost, VIMTS employs a visual MAE-base with around 3 GCN layers as its backbone. Our hyperparameter experiments show that in general settings, a relatively lightweight configuration is optimal, with no significant benefits from additional complexity. In comparison, ViTST uses a Swin Transformer as the visual backbone and a RoBERTa as the text backbone, and requires additional computational cost from image construction, leading to considerable overall costs. The quantified experimental results shown in Table 6 further validate our claims.

### A.3. Discussion about the Effectiveness of Self-Supervised Learning

Our self-supervised learning (SSL) strategy effectively adapts the capability of vision pre-training to multi-channel time series data, leading to robust few-shot capability. In detail, with the GCN module, which learns cross-channel dependencies and complements missing values with information from other channels, VIMTS effectively transforms IMTS into regularly sampled multivariate time series (MTS) at the feature level. Consequently, given that mask-reconstruction-based SSL is effective in learning temporal dependencies in MTS, applying it to enhance IMTS modeling after cross-channel complementation is justified. Furthermore, as the vision pre-training is initially performed on 3-channel data (R-G-B), our SSL strategy is crucial for adapting the model to more diverse application scenarios with a greater number of channels. Our ablation study and few-shot experiments clearly demonstrate a trend: by learning from domain-specific time-channel contexts through SSL, our model can effectively generalize from the 3-channel pre-trained state to handle more varied, complex, and limited data.

### A.4. Analysis of Fine-Tuning Strategies

*Table 8.* Comparison of Different Finetune Strategies. '*' denotes a variant of the existing strategy.

| Finetune | PhysioNet | | Human Activity | | USHCN | | MIMIC | |
|---|---|---|---|---|---|---|---|---|
| | MSE$\times 10^{-3}$ | MAE$\times 10^{-2}$ | MSE$\times 10^{-3}$ | MAE$\times 10^{-2}$ | MSE$\times 10^{-3}$ | MAE$\times 10^{-2}$ | MSE$\times 10^{-2}$ | MAE$\times 10^{-2}$ |
| ALL | $4.95 \pm 0.06$ | $3.53 \pm 0.06$ | $2.69 \pm 0.01$ | $3.31 \pm 0.01$ | $4.91 \pm 0.04$ | $2.98 \pm 0.15$ | $1.39 \pm 0.02$ | $6.59 \pm 0.09$ |
| Attn | $4.91 \pm 0.04$ | $3.55 \pm 0.06$ | $2.67 \pm 0.01$ | $3.12 \pm 0.01$ | $4.90 \pm 0.06$ | $2.91 \pm 0.03$ | $1.41 \pm 0.01$ | $6.58 \pm 0.09$ |
| Bias | $4.86 \pm 0.05$ | $3.58 \pm 0.03$ | $2.65 \pm 0.01$ | $3.08 \pm 0.01$ | $4.85 \pm 0.01$ | $2.96 \pm 0.06$ | $1.40 \pm 0.01$ | $6.58 \pm 0.08$ |
| Frezze | $4.98 \pm 0.06$ | $3.64 \pm 0.03$ | $2.65 \pm 0.01$ | $3.08 \pm 0.01$ | $4.82 \pm 0.01$ | $2.90 \pm 0.05$ | $1.43 \pm 0.01$ | $6.65 \pm 0.11$ |
| MLP | $4.91 \pm 0.03$ | $3.51 \pm 0.04$ | $2.68 \pm 0.01$ | $3.13 \pm 0.01$ | $4.92 \pm 0.04$ | $2.99 \pm 0.12$ | $1.39 \pm 0.01$ | $6.44 \pm 0.04$ |
| Norm | $4.81 \pm 0.07$ | $3.54 \pm 0.04$ | $2.65 \pm 0.01$ | $3.08 \pm 0.01$ | $4.86 \pm 0.02$ | $2.98 \pm 0.05$ | $1.36 \pm 0.02^*$ | $6.40 \pm 0.17^*$ |

To identify the optimal fine-tuning strategy for maximizing the potential of our architecture, we systematically evaluate three distinct approaches, with the results demonstrated in Table 8. (1) **ALL**: Updates all model parameters; (2) **Freeze**:

Retains the pre-trained MAE and the GCN module, and optimizes the remaining parts; (3) Partial Tuning: Selectively updates specific components within the retained parts of **Freeze**, including **Attn** (attention layers), **Bias** (bias terms), **MLP** (MLP blocks), and **Norm** (normalization layers). A variant of **Norm**, marked with ''*, includes GCN, normalization layers, position embeddings, and patch projection layers, and is adapted for datasets with a greater number of channels.

Experiments on three IMTS datasets (PhysioNet, Human Activity, USHCN) reveal that fine-tuning solely the normalization layers (**Norm**) achieves the best overall results, offering an optimal balance in performance across datasets and metrics. Therefore, we have chosen the **Norm** strategy as our default fine-tuning method, which allows for efficient adaptation and maintains the semantic sparse modeling capabilities acquired from visual pre-training. For the MIMIC dataset, which has a larger number of channels, we utilize the variant of **Norm** marked with '*'. This adjustment enables the capture of more intricate cross-channel dependencies while retaining the advantages of the standard **Norm** strategy. Comparisons of various tuning strategies on the MIMIC dataset confirm that this approach delivers the best overall performance.

### A.5. Hyperparameter Sensitivity

We analyze the sensitivity of critical hyperparameters: hidden dimension, patch size, mask ratio, GCN layer depth, and time embeddings dimenson in TTCN (TE) and graph vertex embeddings dimenson in GCN (VE), on all four datasets (PhysioNet, Human Activity, USHCN, MIMIC). We vary each parameter's value while fixing others to their optimal settings derived from preliminary experiments.

**Hidden Dimension.** We test different hidden dimension to balance model performance and computational efficiency. Observations across all three datasets (PhysioNet, Human Activity, USHCN, MIMIC) in Fig. 5 indicate that a hidden dimension size of 32 yields the optimal results for Physionet and USHCN, 64 for Human Activity, while 40 for MIMIC.

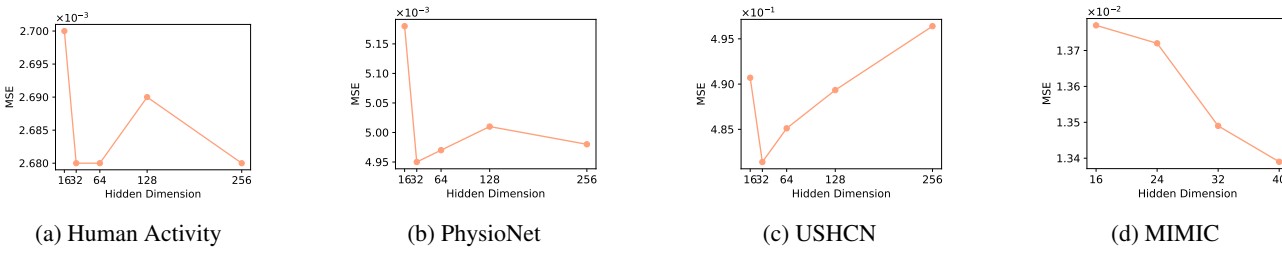

(a) Human Activity  (b) PhysioNet  (c) USHCN  (d) MIMIC

*Figure 5.* Sensitivity of Hidden Dimension

**Patch Size.** As shown in Fig.6, we evaluate patch sizes to find the optimal temporal granularity for each dataset. Too small sizes lack sufficient information due to data sparsity and may cause memory issues, while too large sizes miss fine-grained temporal changes. Based on the results, we select a patch size of 300 (time steps) for Human Activity, 8 for PhysioNet and MIMIC, and 1 for USHCN.

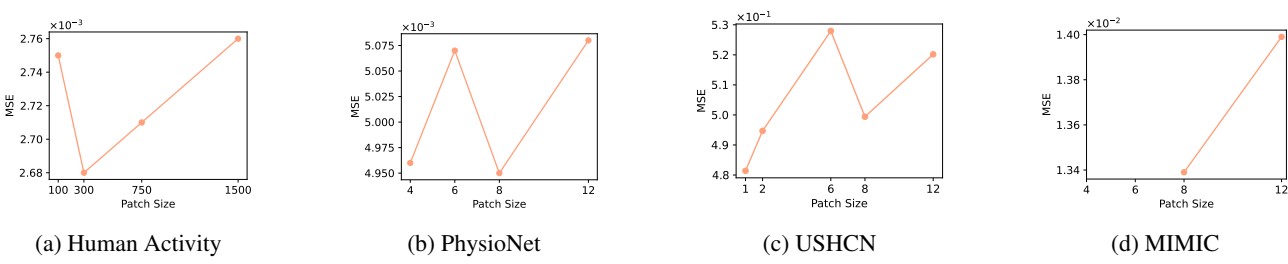

(a) Human Activity  (b) PhysioNet  (c) USHCN  (d) MIMIC

*Figure 6.* Sensitivity of Patch Size

**Mask Ratio.** During self-supervised learning, we vary mask ratios from 0.1 to 0.9 in Fig. 7. For the Human Activity dataset, a ratio of 0.7 provides the best performance. The PhysioNet dataset, which is larger and faces out-of-memory issues, benefits most from a ratio of 0.6. For USHCN and MIMIC, which is larger than PhysioNet, the ratio of 0.4 is optimal.

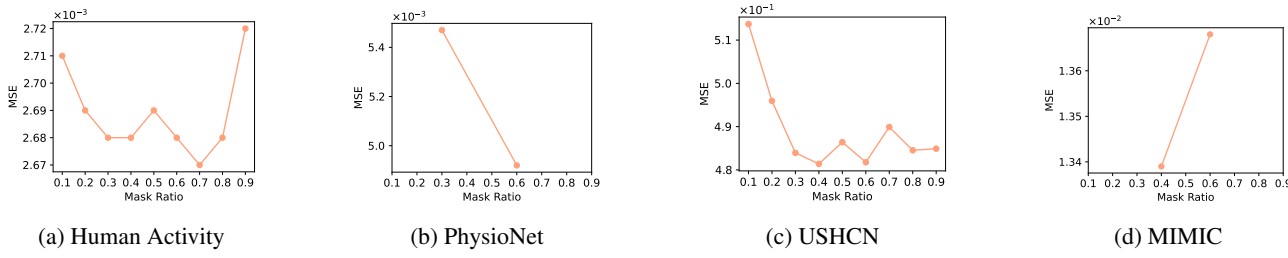

*Figure 7.* Sensitivity of Mask Ratio

**GCN Layer Depth.** Testing GCN layers from 1 to 5, we find that the optimal depths are 2 for the Human Activity dataset, 3 for PhysioNet, USHCN and MIMIC, in Fig. 8. These configurations provide the best balance between model complexity and performance, ensuring effective learning without unnecessary computational overhead and risking overfitting.

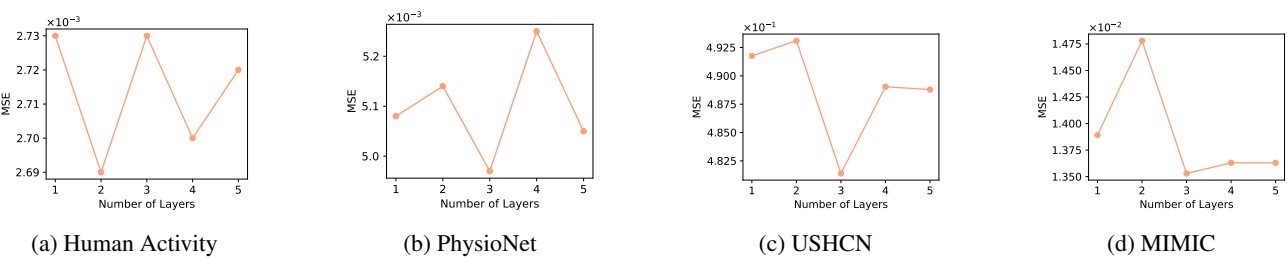

*Figure 8.* Sensitivity of GCN Layer Depth

**TE and VE Dimension.** For effective time × channel feature extraction, as shown in Fig. 9, we test different time TE and VE dimension. For Human Activity and PhysioNet, 5 is optimal. For USHCN, 10 is the best. And for MIMIC, 40 is the best.

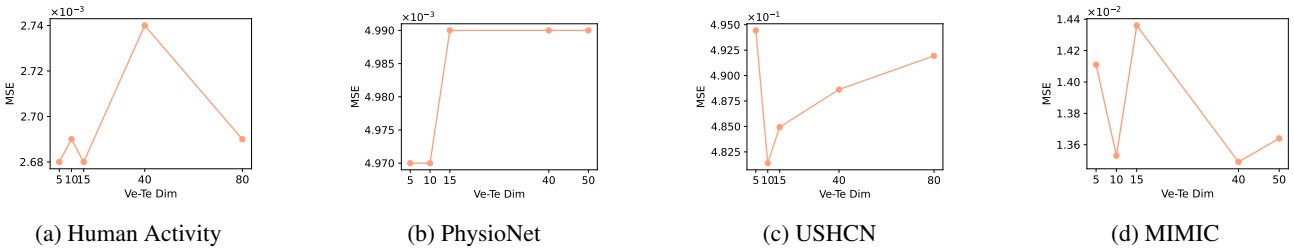

*Figure 9.* Sensitivity of TE and VE Dimension

## B. Baseline

The performances of the models marked with '†' are reported from (Zhang et al., 2024a), as they share the same task setting, evaluation protocols, and datasets split ratios.

### B.1. Explanation of Results on MIMIC Dataset

We identified issues in t-PatchGNN's preprocessing pipeline for MIMIC (refer to their GitHub issues), thus failing to reproduce their results. Utilizing the same configuration as the original paper, after re-evaluating with corrected preprocessing in normal and few-shot settings, VIMTS exhibits competitive MSE, superior MAE compared to t-PatchGNN, and robust few-shot capability. This validates that VIMTS is able to effectively scale to complex and real IMTS data.

### B.2. MTS Forecasting

**DLinear†** (Zeng et al., 2023) decomposes the time series into trend and remainder components using a moving average kernel, then applies two single-layer linear networks to model each component for forecasting. This approach enhances prediction performance on data with clear trends by explicitly handling the trend component.

**TimesNet†** (Wu et al., 2022) introduces a novel approach for time series analysis by transforming 1D time series into 2D tensors to capture both intra-period and inter-period variations, thereby enhancing representation capability. It utilizes TimesBlock, a task-general backbone featuring a parameter-efficient inception block, which can adaptively discover multi-periodicity and extract complex temporal dynamics from the transformed 2D tensors for improved forecasting accuracy.

**PatchTST†** (Nie et al., 2022) is a Transformer-based model that segments time series into subseries-level patches as input tokens and employs channel independence, allowing each univariate time series to share the same embeddings and Transformer weights. This design retains local semantic information, significantly reduces computational and memory demands, and enables the model to consider a longer history.

**Crossformer†** (Zhang & Yan, 2023) is a Transformer-based model designed to address multivariate time series (MTS) forecasting by effectively capturing both cross-time and cross-dimension dependencies. It utilizes a Dimension-Segment-Wise (DSW) embedding to preserve time and dimension information, followed by a Two-Stage Attention (TSA) layer to model these dependencies efficiently. Through its Hierarchical Encoder-Decoder (HED) structure, Crossformer integrates information at various scales for enhanced forecasting performance.

**Graph Wavenet†** (Wu et al., 2019) is a CNN-based method that utilizes a self-adaptive adjacency matrix, learned through end-to-end supervised training, to capture hidden spatial dependencies in graph data. It employs stacked dilated causal convolutions to efficiently model long-range temporal dependencies with an exponentially growing receptive field. This enables Graph WaveNet to effectively handle spatial-temporal graph data for forecasting, combining cross-dimension and cross-time dependency modeling with a gated mechanism.

**MTGNN†** (Wu et al., 2020) tackles spatial and temporal dependencies through a novel graph learning layer, a graph convolution module, and a temporal convolution module. It extracts a sparse graph adjacency matrix adaptively based on data to address spatial dependencies, specifically designed for directed graphs to avoid over-smoothing. The temporal convolution module uses modified 1D convolutions to discover temporal patterns with multiple frequencies and handle very long sequences, effectively capturing cross-dimensional relationships and cross-temporal dependencies.

**StemGNN†** (Cao et al., 2020) is designed to model intra-series temporal patterns and inter-series correlations by transforming multivariate time-series data into the spectral domain using Graph Fourier Transform (GFT) and Discrete Fourier Transform (DFT). This approach enables clearer pattern recognition and more effective predictions by converting structural multivariate inputs into orthogonal time-series representations and then further into frequency domain representations.

**CrossGNN†** (Huang et al., 2023) models MTS forecasting by constructing multi-scale time series with varying noise levels using an Adaptive Multi-Scale Identifier (AMSI). It then applies a cross-scale GNN to capture dependencies between different scales and a cross-variable GNN to handle homogeneity and heterogeneity among variables, using positive and negative edge weights. By focusing on high-saliency edges, CrossGNN achieves linear complexity.

**FourierGNN†** (Yi et al., 2024) improves MTS forecasting by transforming features into Fourier space to handle large-scale graphs more efficiently. Using Fourier Graph Operators (FGO) instead of traditional graph operations, it performs matrix

multiplications in Fourier space, achieving log-linear complexity and high expressiveness. Stacking FGO layers enables effective pattern capture with reduced computational load. Theoretical analysis confirms FGO's equivalence to time-domain graph convolutions.

### B.3. IMTS Classification

**GRU-D**† (Che et al., 2018) is a GRU-based model designed to handle irregularly sampled time series by incorporating representations of missing data patterns through masking and time intervals. Masking indicates which inputs are observed or missing, while time intervals, enhanced with a decay term, capture the patterns of input observations.

**SeFT**† (Horn et al., 2020) reimagines time series classification by treating time series as a set of observations, bypassing the need for ordered sequence processing, which can be disadvantageous in scenarios with irregular sampling or unsynchronized measurements. SEFT leverages advanced set function learning to classify unaligned and irregularly sampled time series, offering improved classification performance and scalability.

**RainDrop**† (Zhang et al., 2022) is a graph neural network designed to model the temporal dynamics and evolving relationships of sensor dependencies in irregularly sampled multivariate time series. By leveraging neural message passing and temporal self-attention, RAINDROP adapts to cross-sample shared relationships between sensors and dynamically estimates unaligned observations, improving the accuracy and interpretability of predictions.

**Warpformer**† (Zhang et al., 2023) is a Transformer-based model that handles intra-series irregularities and inter-series discrepancies by using a specialized input representation. This encoding captures signal values, sampling times, and intervals. A warping module then aligns all series to a unified scale through down-sampling or up-sampling. Subsequently, a doubly self-attention module processes the synchronized data for enhanced representation learning, improving the handling of irregular time series and predictive performance.

### B.4. IMTS Interpolation

**mTAND**† (Shukla & Marlin, 2021a) handle multivariate, sparse, and irregular time series using a continuous-time approach with learned embeddings and a time attention mechanism. This model re-represents time series data at fixed reference points, using an encoder to convert irregular inputs into fixed-length latent representations and a decoder for reconstruction or forecasting. By replacing fixed similarity kernels, mTANs offer greater flexibility and can be adapted for forecasting by modifying the interpolation queries.

### B.5. IMTS Forecasting

**Latent-ODE**† (Rubanova et al., 2019) enhances traditional RNNs by modeling continuous-time dynamics with neural ODEs. It serves as both a standalone autoregressive model and a recognition network in the Latent ODE model, which evolves an initial latent state over time for generating time series data. This approach integrates continuous-time dynamics into RNNs, offering better management of continuous-time data and more expressive temporal patterns.

**CRU**† (Schirmer et al., 2022) is a probabilistic architecture for modeling irregularly sampled time series, mapping observations into a latent space governed by a linear SDE. Using the Kalman filter's continuous-discrete formulation, CRU propagates latent states and integrates new observations. This approach provides explicit uncertainty estimates, ensures optimal state updates in locally linear spaces, and enables analytical resolution of latent states, thus avoiding numerical integration.

**Neural Flow**† (Biloš et al., 2021) is a neural network approach that directly models the solution curves of ordinary differential equations (ODEs), eliminating the need for expensive numerical solvers required in traditional methods. By designing flow architectures that meet specific conditions, the model significantly improves computational efficiency while retaining the modeling capabilities of neural ODEs.

**t-PatchGNN**† (Zhang et al., 2024a) transforms univariate irregular time series into transformable patches with a unified time horizon, bypassing pre-alignment and capturing richer local semantics. This approach aligns IMTS in a consistent temporal resolution, addressing asynchrony issues. It uses a time-aware convolution network to encode patches into latent embeddings, which are processed by a Transformer for intra-series dependencies. Time-adaptive graph neural networks then model inter-series correlations through dynamic graphs, with a final MLP layer generating forecasts based on the comprehensive latent representation.

**VisionTS** (Chen et al., 2025) explores the use of pre-trained visual models for time series forecasting (TSF) by interpreting pixel variations in images as temporal sequences. This approach leverages the similarities between images and time series, such as their continuous nature, real-world origins, information density, and shared features. Focusing on a visual masked autoencoder (MAE), a popular computer vision model, VisualTS reformulates TSF as a patch-level image reconstruction task. By transforming 1D time series data into 2D matrices and rendering them as images, the method aligns the forecasting window with masked image patches, enabling zero-shot forecasting without further adaptation. This innovative approach bridges the gap between pre-training on images and downstream TSF tasks, offering a promising direction for leveraging visual models in TSF. During training, it applies a learning rate of $1e^{-4}$ and and aligns the input sequences to a uniform temporal grid via linear or zero interpolation, with an interpolation resolution of 30 points per patch for the output.

