# OpenReview forum: "IMTS is Worth Time $\times$ Channel Patches: Visual Masked Autoencoders for Irregular Multivariate Time Series Prediction"
_ICML.cc/2025/Conference — ICML 2025 poster_

### Official Review · Reviewer_rRas · 2025-02-23

**Overall Recommendation:** 2

**Summary:**

This paper addresses forecasting on Irregular Multivariate Time Series (IMTS), where observation intervals are variable and there are missing values. The authors propose to use Masked AutoEncoder (MAE) to efficiently handle missing values in IMTS and scale pre-training/fine-tuning. They also propose to use a Graph Convolutional Neural Network (GCNN) for handling a variable number of channels in different time series, temporal period embedding for handling timestamps as a variable in the MAE framework, and the coarse-to-fine strategy for focusing on relevant temporal context in the forecasting of the specific timestamp. Experimental results on multiple datasets demonstrated that the proposed method performs better than the existing methods.

**Claims And Evidence:**

* Eq.6 is not justified well. It should be discussed in literature or theoretically.
* The usage of GCN in Section 2.3.3. is not justified well. It should be discussed in literature or theoretically.
* Eqs.11-15 for the time period embeddings are not justified well. It should be discussed in literature or theoretically.
* Similar comments to the above for Sections 2.4.2. and 2.5.

**Essential References Not Discussed:**

NA

**Experimental Designs Or Analyses:**

I checked Section 3 and the appendices.

**Methods And Evaluation Criteria:**

Make sense.

**Other Comments Or Suggestions:**

NA

**Other Strengths And Weaknesses:**

The proposed method may have novel points and have some practical impact, but the clarity issues are too severe to understand the method.
* We tend to use "forecasting in this problem setting rather than "prediction."
* Figure 1 is not referred to in the main text.
* Figure 2 is hard to understand. Where are the results of DLinear and VisionTS?
* In Section 2.2, L, N, and q^n_j are not defined.
* Paragraphs "IMTS Representation" and "IMTS Prediction" look to share some portion of notations, but their relationship and dependencies are not explicitly described, which makes it hard to follow Section 2.2.
* The relationship between t_start^p, l_p, and r_p is unclear.
* In Eqs.2, 5, and 7, "||" might be some operator, but it is not defined.
* In Eq.3 and the description under Eq.3, L_p and D_in are not defined (L_p can be s, maybe?). The character d may be used in different meanings from Eq.1. And, what is the meta-filter?
* In Eq.4, the superscript "*" is not defined.
* In Eq.4, can the subscript l_p:r_p be just i and remove [i]?
* In Eq.5, the variable m is used inconsistently with subscript and without subscript.
* Eqs.7-9 are hard to understand. What is the superscript s? Why k is in {1,2}
* The equation in l.167 of the right-hand side of p.4 is incorrect.
* In the equation in l.180 of the right-hand side of p.4, R^2D should be R^Px2D.
* Eqs.11-15 are hard to follow.
* In Eq.20, TPE does not require the projection into the input dimension of the decoder?
* In Eq.21, the variable e is used inconsistently from the previous formulation.
* The losses in Eqs.26 and 27, operation (summation or averaging) over n lacks.
* In l.312 of the left-hand side of p.6, it is better to use consistent notations with the main text (x should be used, not y).
* Tables 1 and 2 appear in an inverted order.

**Questions For Authors:**

See above.

**Relation To Broader Scientific Literature:**

The usage of MAE for time series forecasting with missing values is novel.

**Theoretical Claims:**

NA

---

> ### Author Rebuttal · Authors · 2025-04-01
>
> We thank the reviewer for their constructive feedback and recognition of our work’s **novelty and practical impact**, addressing each point below for clarity.
> ## Overview of Method
> VIMTS processes IMTS by: 1) segmenting into fixed intervals with variable-length points per channel; 2) extracting unified patch representations via dynamic convolution; 3) compensating missing values via GCN capturing inter-channel dependencies; 4) modeling temporal patterns across P×N patches using MAE; 5) pre-training with masked patch reconstruction to transfer MAE’s sparse processing, then fine-tuning; 6) coarse-to-fine decoding (patch reconstruction → timestamp queries via continuous embeddings). This vision-inspired pipeline uniquely adapts to irregularity-aware forecasting. Notations will be clarified in revision.
> ## Methodological Clarifications
> ### Q1: Justification of Channel Embeddings (Eq.6)
> Learnable channel embeddings $e_n$ model channel-specific traits (units, stats, missing patterns), inspired by positional and cross-modal retrieval’s modality embeddings [3]. They distinguish heterogeneous channels (e.g., physiological vs. environmental) during cross-channel compensation and temporal modeling.
> ### Q2: GCN Usage in Cross-Channel Interaction (Sec 2.3.3)
> Our GCN addresses sparse/misaligned observations by modeling bidirectional channel dependencies. We fuse static embeddings (inflow/outflow nodes) with dynamic patch features via gated fusion: $g_{p,k} = \text{ReLU}\left(\tanh\left([H_p \parallel E_k^s]W_k^g\right)\right),$to obtain hybrid embeddings $E_{p,k}$. The adaptive adjacency matrix $A_p = \text{Softmax}(\text{ReLU}(E_{p,1}E_{p,2}^\top))$ dynamically captures directional relationships, enabling context-aware message passing. Ablations (Tab 3) confirm GCN’s necessity in compensating missing values through cross-channel interactions, aligning with evolving graph structures in spatio-temporal GNNs [1,2].
> ### Q3: Time Period Embeddings (Eq.11-15)
> Our TPE adapts ViT’s 2D positional encoding [4] for temporal data: horizontal axis encodes patch order $p$, vertical axis fixes duration $s$. Initialized with sinusoidal functions, it preserves MAE’s positional awareness while adapting to IMTS patterns during fine-tuning. This dual-axis design jointly maintains sequential order and temporal scale information.
> ### Q4: MAE Reconstruction Mechanism (Sec.2.4.2)
> MAE reconstruction masks $r%$ patches (e.g., 60% for PhysioNet) to simulate missing segments. The model infers missing regions via neighboring patches and cross-channel dependencies. The decoder fuses visible tokens $z_p$ and mask tokens $[M]$ (projected via $W_{dec}$) with TPE ensuring temporal coherence, extending MAE’s sparse processing [6] to irregular series through patch-level masking.
> ### Q5: Patch-to-Point Prediction (Sec.2.5)
> Our coarse-to-fine strategy bridges patch-level semantics and timestamp specific queries: (1) MAE decoder reconstructs dense patch representation $ẑ_{i_q}^m$ aggregating temporal patterns and cross-channel contexts. (2) Continuous-time embedding $\phi(t_q)$ injects fine-grained temporal context. (3) An MLP fuses $ẑ_{i_q}^m$ and $\phi(t_q)$ for final prediction $\hat{x}_q$. This hierarchical approach mirrors image super-resolution (low-resolution features guide high-detail reconstruction) adapted for temporal irregularity via timestamp embeddings.
> ## Technical Corrections
> - Notation
>   - $L$: total timestamps; $N$: channels; $q_j^n$: channel-specific queries;  unified $\mathcal{O} = (\mathcal{T}, \mathcal{X}, \mathcal{M})$
>   -  $t_\text{start}^p$: patch. start; $l_p/r_p$: first/last timestamp index
>   - "$\|$": feature concatenation
> - Equation Fixes
>   - Eq.5 : $m_p → m_p^n$ (channel-specific mask)
>   - Eq.20: Dimension $\mathbb{R}^{P \times 2D}$
>   - Losses (Eqs.26-27): Normalized by ($|\mathcal{Q}|$)
> - Figs/Tabs
>   - Fig 1 : Added contrast with prior work
>   - Fig 2: Adjusted DLinear/VisionTS scales
>   - Tab 1 & 2: Order corrected.
> ## Reviewer-Specific Queries
> ### Q6 (Eq.167)
> Fixed as $\mathbf{H}'_p = [\mathbf{H}_p \| \mathbf{H}_p^\text{gcn}] \in \mathbb{R}^{N \times 2D}$.
> ### Q7 (Eq.3)
> Meta-filter = MLP generating adaptive convolution kernels [5] for variable-length inputs.
> ### Q8 ($k$ in Eqs.7-9)
> $k \in \{1,2\}$ denotes inflow/outflow embeddings for bidirectional dependencies.
> ### Q9 (Loss)
> Loss averages over queries ($|\mathcal{Q}|$), not channels, due to timestamp misalignment.
>
> [1] Graph wavenet for deep spatial-temporal graph modeling. Arxiv.
>
> [2] Irregular multivariate time series forecasting: A transformable patching graph neural networks approach. ICML24.
>
> [3] Cross-modality transformer for visible-infrared person re-identification. ECCV22.
>
> [4] An image is worth 16x16 words: Transformers for image recognition at scale. Arxiv.
>
> [5] Irregular traffic time series forecasting based on asynchronous spatio-temporal graph convolutional networks. KDD24.
>
> [6] Masked autoencoders are scalable vision learners. CVPR22.

---

### Official Review · Reviewer_vVGF · 2025-03-05

**Overall Recommendation:** 3

**Summary:**

The paper introduces VIMTS, a framework adapting MAE for IMTS prediction, addressing challenges like unaligned signals and missing values. Unlike existing methods that separately model temporal and channel patterns, VIMTS enhances representation learning by transforming sparse signals into image-like patches, capturing cross-channel dependencies. The model leverages MAE’s pre-trained capability for patch reconstruction and refines predictions with a coarse-to-fine approach. Additionally, it integrates self-supervised learning with supervised fine-tuning. Extensive experiments demonstrate VIMTS’s performance and few-shot capability, expanding the application of visual foundation models to time series prediction.

-------Reply after rebuttal:
Thank you for the response, it addresses most of my concerns. However, after considering multiple factors, I’ve decided to maintain my original score.

**Claims And Evidence:**

The claims in line 23 (left) and lines 29-30 (right) on page 1 stating that existing pre-training models are limited to UTS are inaccurate, as many existing studies have already considered multivariate time series.

**Essential References Not Discussed:**

Li Z, Li S, Yan X. Time series as images: Vision transformer for irregularly sampled time series[J]. Advances in Neural Information Processing Systems, 2023, 36: 49187-49204.

**Experimental Designs Or Analyses:**

The experiments in the paper are sufficient, with appropriate comparison methods and thorough analysis. The use of commonly employed real-world datasets enhances the credibility of the results.

**Methods And Evaluation Criteria:**

The model designed in the paper is both reasonable and effective for addressing the IMTS modeling problem. The dataset used is a commonly employed real-world dataset in IMTS modeling, which also provides practical guidance for solving real-world problems.

**Other Comments Or Suggestions:**

1. **Figure 2** contains too many elements, making the focus unclear. It would be beneficial to simplify the figure by selecting key components that highlight the advantages of the proposed method.
2. In **Figure 3**, the legend is only represented by letters, which is not sufficiently clear. It is recommended to add descriptive text, such as "Channel Mask", to improve clarity.

**Other Strengths And Weaknesses:**

1. The authors should conduct a more in-depth comparison and analysis between this work and **ViTST [1]**. If the authors choose not to include this comparison, the difference in tasks should not be used as an excuse, as both models first learn representations and then use a projection head for downstream tasks.
2. The contributions section is overly lengthy.
3. Since the model incorporates multiple GCN layers, it would be helpful for the authors to compare the **time and space complexity** of the proposed method with other SOTA models.

[1] Li Z, Li S, Yan X. Time series as images: Vision transformer for irregularly sampled time series[J]. Advances in Neural Information Processing Systems, 2023, 36: 49187-49204.

**Questions For Authors:**

See Other Strengths And Weaknesses.

**Relation To Broader Scientific Literature:**

This paper details efforts to advance IMTS forecasting. The paper provides a new perspective on the forecasting problem of IMTS to some extent and effectively improves forecasting performance.

**Theoretical Claims:**

There are no theoretical claims in this paper.

---

> ### Author Rebuttal · Authors · 2025-04-01
>
> We sincerely appreciate your recognition of the practical significance and our innovation for improving IMTS forecasting performance, as well as the insightful feedbacks. Below, we address each concern raised by the reviewer in detail:
>
> ## Q1: Incorrect claims regarding pre-trained models on MTS
>
> We will clarify in the revision that our claim applies specifically to IMTS forecasting scenarios, while most existing pre-trained model based forecasting methods are for regular sampled time series data.
>
> ## Q2. Comparison with ViTST
>
> We appreciate the reviewer's suggestion. Although both ViTST [1] and VIMTS both employ image-like representations for time series, there are key distinctions:
> - **Differences in Innovation Points**: To extract information from IMTS data, **ViTST** deal with missingness with **interpolation and transformation from IMTS to images**, which causes **information loss with computational overhead**, makes inputs less precise for understanding patterns in data, failing to perform well in forecasting.
>
>   In contrast, **our model divides data by time intervals, patchifying to time x channel patches and extracts block features** without interpolation, which preserves **data integrity** and creates more precise inputs for MAE  to model internal data structures without computational overhead of image construction.  Further, it explicitly models inter-channel interaction with GCN, compensating for missingness across channels.
>
>   Our analysis is further supported by empirical analysis, which evaluates the model performance onPhysioNet.
>
> |Method| MSE($10^{-3}$) | MAE($10^{-2}$) |
> | - | - | -|
> | VIMTS (Ours)| 4.81 ± 0.07| 3.54 ± 0.04|
> | ViTST| 66.37 ± 0.08| 20.16 ± 0.05|
> | VisionTS (Zero interpolation) | 42.41 ± 0.02 | 13.13 ± 0.02 |
>
>
>   Note that this experiment also involves VisionTS [2] with similar image-based methods, and confirms that despite its powerful visual MAE framework and strong performance on regular time series, it similarly deteriorates when processing irregular data.
>
> - **Computational Cost**: **VIMTS** employs a visual MAE backbone (base) with limited number (around 3) of GCN layers. Our hyperparameter experiments show that a lightweight configuration (node features - 5 ~ 10 dimension, and projectors - 32 * 32 ~ 64 * 64) is optimal, with no significant benefits from additional complexity.
>
>   In comparison, **ViTST** uses a Swin Transformer as the visual backbone and a Robert as the text backbone, resulting in considerable computational costs, the quantified experimental results shown in the table in Q3 further validate our claims.
>
> ## Q3. Time and space complexity analysis
>
> We acknowledge the importance of complexity analysis, and will include the table below comparing parameters and inference times in the revision.
>
> | | VIMTS(base)| VIMTS(base) w/o GCN  | ViTST   | tPatchGNN | CRU  | WrapFormer |
> | - | - | - | - | - | - | - |
> | **Trainable Param**| Pretrain: 111.01M Finetune: 332.61k | Pretrain: 110.98M Finetune: 332.61k | 212.31M | 943.99k  | 28.65k  | 178.06K   |
> | **Avg Training Time (s) / epoch**  | 87.80   | 84.96 | 140.20   | 17.14 | 172.45 | 22.67 |
> | **Avg Inference Time (ms) / instance** | 4.82  | 4.34  | 7.09   | 1.60 | 7.66   | 5.48  |
>
> **Parameter Numbers**: VIMTS utilizes visual MAE-base as backbone, which can be finetuned on a single RTX 4090, the GCN layers in VIMTS are lightweight, contributing only around 32.4k parameters. The number of trainable parameters is acceptable in real-world application and lower than other vision foundation model based methods like ViTST.
>
> **Time Complexity**: VIMTS achieves efficient training (87.8s/epoch) while its inference speed (4.815ms/instance) outperforms ViTST (visual foundation model - 7.094ms/instance), CRU (Neural-ODE - 7.655ms/instance), and WrapFormer (Transformer - 5.475ms/instance), making it practical for real-world deployment. Though it's slightly slower than lightweight models like tPatchGNN (1.604ms), this trade-off is justified by superior accuracy and improved data efficiency from pre-training—critical in medical applications where sub-5ms latency remains acceptable.
>
> ## Q4. Revisions for better readability
>
> As for Figure 2, we will revise it by selecting representative baseline methods, adjusting the scale and using a clearer color scheme to highlights the model's superior performance and few-shot ability.
>
> As for Figure 3, we will add more intuitive and descriptive text to the legends in the figure to enhance the readability.
>
> As for the Contribution Section, we will streamline it in revision to focus on our core insights and make it more concise.
>
> ## Reference
>
> [1] Time series as images: Vision transformer for irregularly sampled time series, NIPS23
>
> [2] Visionts: Visual masked autoencoders are free-lunch zero-shot time series forecasters, arXiv:2408.17253.

---

### Official Review · Reviewer_kMse · 2025-03-13

**Overall Recommendation:** 3

**Summary:**

This paper presents VIMTS, which exploits the ability of visually pre-trained MAEs to model semantically sparse multi-channel data for IMTS prediction. Specifically, IMTS data is treated as image-like patches across temporal and channel dimensions during the encoding process, which are divided into equally spaced patches where TTCN extracts in-channel features and GCN aggregates cross-channel information to complement the patch representation with missing values. For decoding, VIMTS employs a coarse-to-fine technique that first generates patch-level predictions and then refines them into precise time-point predictions. Extensive experiments on real datasets validate the effectiveness of VIMTS.
## update after rebuttal
The author's response has solved some of my confusion and I keep my score.

**Claims And Evidence:**

yes

**Essential References Not Discussed:**

No

**Experimental Designs Or Analyses:**

yes

**Methods And Evaluation Criteria:**

yes

**Other Comments Or Suggestions:**

No

**Other Strengths And Weaknesses:**

Strengths：
1, VIMTS performs well. Compared to previous methods such as T-PATCHGNN, VIMTS shows better prediction performance on three datasets.

2, The methodology is novel. Inspired by VAE, VIMTS combines supervised fine-tuning with self-supervised learning to enhance IMTS modelling.

Weaknesses:

1, Figure 3 is difficult to understand, what does Lssl, Lft, Φ mean? Suggest adding figure notes.
2，The datasets listed in Table 1 all have a relatively small number of variables, which does not adequately validate the effectiveness of self-supervised training in irregular multivariate time series forecasting tasks. Some datasets with more number of variables are suggested to be added. For example, the MIMIC dataset.
3，

**Questions For Authors:**

No

**Relation To Broader Scientific Literature:**

This paper is inspired by the multi-channel semantic sparse information modelling capability and time series domain adaptation of visual MAE, which effectively improves IMTS prediction performance.

**Theoretical Claims:**

yes

---

> ### Author Rebuttal · Authors · 2025-04-01
>
> We sincerely appreciate your recognition of the **novelty** and our efforts to leverage the **multi-channel semantic sparse information modeling capability** and **time series domain adaption** of visual MAE. Regarding the ambiguity you pointed out and your other concerns, we address them below.
>
> ## Q1: Revisions to notations in Figure 3
>
> We appreciate your careful review about the ambiguity in the notations used in Figure 3. To clarify the specific notations you mentioned:
> - $L_{ssl}$: This represents the L1 loss value calculated during the self-supervised learning phase.
> - $L_{ft}$: This denotes the L1 loss value calculated during the fine-tuning phase.
> - $Φ(t)$: This represents the time embedding for a given timestamp t, with the explicit formula provided in Equation (1), we utilize it to generate precise prediction from the matching time period and channel representations.
> We will incorporate these detailed explanations into the caption of Figure 3 to enhance its clarity. Furthermore, we will include a comprehensive notation summary table within the appendix to improve overall readability.
>
> ## Q2: Suggestions about validating the effectiveness of self-supervised learning
>
> Thank you for highlighting the importance to validate the **effectiveness of self-supervised learning**. Our framework inherently benefits from multi-channel datasets. Our self-supervised learning (SSL) strategy effectively **adapts the vision pre-trained model to multi-channel time series data**, enabling our model to more **efficiently extract information from different time intervals** and improve performance on forecasting tasks. We will address this issue from both theoretical and empirical perspectives.
>
> **Theoretical Analysis**:
> With the **GCN module**, which captures inter-channel interactions and complements missing values, VIMTS effectively transforming the IMTS problem into a regular sampled multivariate time series (MTS) problem. Given that MAE-based SSL is effective in learning temporal structures under complete data conditions, thus our SSL framework can leverage these representation to enhance the IMTS modeling. This theoretical advantage leads to improved performance on multi-channel datasets like MIMIC.
> Furthermore, as the vision pre-training is initially performed on 3-channel data (**RGB space**), our SSL strategy is crucial for adapting the model to more diverse application scenarios with a **greater number of channels**. By learning from other channels and temporal contexts, our model can effectively generalize from the 3-channel pre-trained state to handle more varied and complex data.
>
> **Empirical Analysis**:
> We sincerely appreciate the reviewer’s suggestion to validate our method on high-dimensional datasets. Following this guidance, we conducted new experiments on the MIMIC dataset (with $96$ channels) and compared against the state-of-the-art t-PatchGNN. Relevant data and a supplementary figure are available in this website:
>
>  **[anonymous repo]**(https://anonymous.4open.science/r/VIMTS-Rebuttal-028D/).
>
> Our ablation study and few-shot experiments clearly demonstrate a trend: with the SSL strategy, our model exhibits significant **data efficiency and strong performance even with limited data**. This supports our argument that SSL is effective in multi-channel scenarios.
>
> Moreover, our results demonstrate that VIMTS achieves superior performance compared to the current state-of-the-art t-PatchGNN, as shown in the table below: (mean of the results from 5 seeds)
>
> | Metric | VIMTS      | t-PatchGNN (Reported) |
> |--------|----|-----------------------|
> | MSE    | $1.3654 \times 10^{-2}$ | $1.69 \times 10^{-2}$     |
> | MAE    | $6.4482 \times 10^{-2}$ | $7.22 \times 10^{-2}$     |
>
> Notably, we identified inconsistencies and bugs in t-PatchGNN’s preprocessing pipeline for MIMIC (refer to their GitHub issues). Upon re-evaluating their model with corrected preprocessing, VIMTS still exhibits competitive MSE (vs. t-PatchGNN’s $0.013608 \pm 0.000179$) and superior MAE (vs. $0.0656 \pm 0.001141$). This validates that VIMTS not only scales effectively to high-dimensional IMTS but also generalizes robustly across varying missingness patterns. These additional experiments further confirm our theoretical analysis: the GCN-enhanced SSL framework more effectively leverages inter-channel correlations, leading to stronger performance in real-world IMTS scenarios. We will include these results in the revised manuscript.

---

> > ### Comment · Reviewer_kMse · 2025-04-08
> >
> > Thanks to the author for the reply. Taking into account my comments and those of the other reviewers, I think that there is a real problem with the readability of the current paper. Therefore I decide to keep my score.

---

> > > ### Author Response · Authors · 2025-04-08
> > >
> > > We sincerely thank you for your valuable feedback and for dedicating time to review our work. Readers familiar with irregular multivariate time series forecasting would likely find our presentation more accessible. Nevertheless, we recognize the importance of clear presentation and will implement your suggestions to enhance figures, refine notations, and improve module descriptions in our revision.
> > >
> > > We believe that with the feedback from you and other reviewers, our revision will be more clear and our contributions will be more effectively communicated to the community.

---

### Official Review · Reviewer_vWeW · 2025-03-14

**Overall Recommendation:** 3

**Summary:**

This paper proposes a new approach for Irregular Multivariate Time Series (IMTS), characterized by unaligned multichannel signals and massive missing values. Instead of modeling temporal and channel patterns in isolation, as in most of current research, this paper proposes VIMTS, a framework that adapts Visual Mask autoencoder (MAE) for IMTS prediction and jointly model temporal-channel patterns. The intuition is that formatting IMTS into image-like patches will utilize cross-modality learning. The core idea is to utilize a pre-trained vision MAE (He et al., 2022) to model time period patches of IMTS data with multichannel information.

**Claims And Evidence:**

The claims that the proposed approach addresses the aforementioned issues are generally well supported. See detailed comments and suggestions in sections below.

**Essential References Not Discussed:**

The literature review of this paper is comprehensive.

**Experimental Designs Or Analyses:**

Experiments are included and results show the proposed method perform well comparing to baselines. The paper also includes a comprehensive ablation study to check the role of each component of the model architecture.
The major comment is that while there are about 20 baselines selected, not all of them are state of the art. In particular, the paper would benefit from comparing to approaches that use the transformer architecture and incorporates the masking approach.

**Methods And Evaluation Criteria:**

The architecture in this paper consists of three main components: time×channel patchify, time-wise reconstruction, and the patch2point Prediction. The authors claim the time*channel patchification is one of the innovations of this paper. However, this approach seems to be similar to the approaches used in other papers, such as
Jungo, J., Xiang, Y., Gashi, S., & Holz, C. (2024). Representation Learning for Wearable-Based Applications in the Case of Missing Data. *Human-Centric Representation Learning workshop at AAAI 2024*.
It is also a natural way to adopt the vision MAE to the time series. Therefore, please clarify the difference of your approach and innovation.

**Other Comments Or Suggestions:**

It would be helpful to discuss the trade-off between the model's performance and computational efficiency, particularly given the multi-step nature of the proposed approach.

**Other Strengths And Weaknesses:**

The notation part could use more detailed definitions. For example, please clarify the definition of notation N and L. While their meaning can be interpreted from the context, it would be more rigorous to formally define them and explain their practical meaning in applications. A thorough check of the notation and the formulae is recommended.

In line 312, the general definitions of MAE and MSE are given. However, it is not clear the definition of $y$ in the context of the tasks of this paper. Please further specify the definition of such errors.

**Questions For Authors:**

None.

**Relation To Broader Scientific Literature:**

Although several studies have explored applying Vision Masked Autoencoder (MAE) methods to time series data, the challenges posed by irregular sampling frequencies and significant missing data remain common and critical, particularly in healthcare applications. This paper addresses these gaps and has good potential for broad application in practical scenarios.

**Theoretical Claims:**

NA

---

> ### Author Rebuttal · Authors · 2025-04-01
>
> We deeply appreciate the reviewers' recognition of our method's **novelty**, **comprehensive experiments**, and **potential for broad application**. We address your concerns below:
>
>
> ## Q1: Originality and differentials from Jungo et al.
>
> While both works explore patchification, our core innovation lies in：
>
> 1.  **Enhanced Channel Dependency Modeling with GCN:** VIMTS employs GCNs to learn both static and dynamic information in a asymmetric manner, therefore model bidirectional inter-channel dependencies, enabling explicit cross-channel compensation for missing values—which is more consistent with the real information dependence situation than Jungo et al.'s projection-only method. Ablation studies verify Its effectiveness.
> 2.  **Leveraging Vision Pretraining:** We utilize vision pretraining capabilities, providing strong initialization for learning sparse patterns from limited data. It's a vital component not explored in Jungo et al.
>
> 3.  **Coarse-to-fine Strategy**: As confirmed in our ablations, VIMTS forecasting with a encoder-decoder architecture, enabling more precise predictions generated from related time-segment and channel contexts, outperforming direct projection the approach like Jungo et al., which generates prediction through global projection features
>
> ## Q2: Comparison to other transformer based baselines with mask reconstruction
>
> Our experiments have included SOTA Mask-Reconstruction (MR) transformer variants like PatchTST and VisionTS, demonstrating VIMTS's superior performance:
>
> **Limitations of Existing Methods on IMTS:** While the transformer with MR like **PatchTST** [3] perform well on regular sampled multivariate time series (MTS) data, they lack explicit mechanisms to handle irregular sampling and inter-channel modeling for missingness complementation. This leading to significant performance drops in IMTS data. Moreover, **VisionTS** [2], a vision foundation model with MR, performs well on MTS but fails to generalize to IMTS tasks due to its reliance on gridding, resizing, and interplotation when processing IMTS. Similarly, **MOMENT** [1], a time series pre-trained foundation model with MR underperforms even PatchTST in MTS forecasting (MOMENT - Table [1]) and cannot address irregular sampling or missing value issues.
>
> **Ablation Study Reinforces VIMTS's Design:** As mentioned in the comparison to Jungo's model, encoder-only architectures struggle with IMTS data even when using self-supervised learning. Our coarse-to-fine prediction strategy proves essential for achieving precise time-segment and channel-specific forecasts.
>
> ## Q3: Revisions of notations and definitions
>
> For improved clarity and technical rigor, we will add a **notation table** in the appendix and better integrate figures/tables throughout the manuscript.
>
> Regarding the ambiguity you've mentioned:
>
> * $N$: total number of channels; $L$: count of unique timestamps.
> * $q_j^n$: $j$-th query timestamp in channel $n$.
> * $y_i$: ground-truth value at $i$-th query timestamp; $\hat{y}_i$: model's prediction.
>
> ## Q4: Model performance and efficiency trade-off
>
> Our self-supervised learning (SSL) effectively offers a **favorable trade-off** with:
>
> * **High Data Efficiency and Few-Shot Performance:** With SSL, VIMTS achieves almost **SOTA** on three IMTS datasets using only **20% of the training data**. This significantly reduces the required training data volume, enabling faster model development and deployment in data-scarce scenarios.
>
> * **Manageable Model Complexity:** Our analysis shows **excellent performance without excessive scaling**, requiring only 2-4 GCN layers and several simple mlp predictor except for the MAE backbone. In more previous trials, scaling MAE beyond the base level did not improve results and caused memory issues, aligning with the case in VisionTS [2], which applies visual MAE to MTS forecasting.
>
> * **Improved Performance with Controllable Computation Cost:** On PhysioNet, our ssl delivers over 10% performance improvement for ~40% additional training time—a worthwhile trade-off for three reasons:
>     1.Our model relies on less training data for the same results, mitigating the data constraints.
>
>     2.Our training of each epoch remains faster than ODE-based and other vision based methods (**Reviewer vVGF Q3**) and is acceptable in real-world applications.
>
>     3.SSL adds no inference cost. VIMTS maintains faster inference than most competitors while staying competitive with t-PatchGNN. Since real-world applications deployment scenarios often face data limitations and prioritize inference efficiency over training costs, and this cost is also acceptable, consequently this trade-off provides substantial practical benefits.
>
> ## Reference
>
> [1] MOMENT: A family of open time-series foundation models, ICML24
>
> [2] Visionts: Visual masked autoencoders are free-lunch zero-shot time series forecasters, arXiv:2408.17253
>
> [3] A Time Series is Worth 64 Words: Long-term Forecasting with Transformers, ICLR23.

---

### Decision · Program_Chairs · 2025-05-01

**Decision:**

Accept (poster)

**Comment:**

This paper introduces VIMTS, a novel framework that adapts visual Masked Autoencoders (MAE) for the prediction of Irregular Multivariate Time Series (IMTS). Extensive experiments highlight VIMTS’s superior performance and robust few-shot capabilities, showcasing its potential to extend visual foundation models to broader time series prediction tasks.

The paper received four reviews. Following the rebuttal and subsequent discussion, three reviewers reached a consensus to accept the paper, citing its innovative contributions, strong empirical performance, and potential broader impact. The fourth reviewer, who issued a weak reject, raised concerns primarily related to the presentation rather than the core methodology. After reviewing the authors' rebuttal, the AC finds that the concerns about presentation have been adequately addressed.

Overall, the AC believes the paper presents valuable contributions and recommends it for acceptance. To further enhance its quality, the AC encourages the authors to revise the manuscript by incorporating all reviewer suggestions—particularly by including the additional experimental results presented during the rebuttal phase and improving the overall presentation.